# Sustained TNF-α stimulation leads to transcriptional memory that greatly enhances signal sensitivity and robustness

Zuodong Zhao[1†], Zhuqiang Zhang[1†], Jingjing Li[1,2], Qiang Dong[1], Jun Xiong[1], Yingfeng Li[1], Mengying Lan[1,2], Gang Li[3], Bing Zhu[1,2]*

[1]National Laboratory of Biomacromolecules, CAS Center for Excellence in Biomacromolecules, Institute of Biophysics, Chinese Academy of Sciences, Beijing, China; [2]College of Life Sciences, University of Chinese Academy of Sciences, Beijing, China; [3]Faculty of Health Sciences, University of Macau, Macau, China

**Abstract** Transcriptional memory allows certain genes to respond to previously experienced signals more robustly. However, whether and how the key proinflammatory cytokine TNF-α mediates transcriptional memory are poorly understood. Using HEK293F cells as a model system, we report that sustained TNF-α stimulation induces transcriptional memory dependent on TET enzymes. The hypomethylated status of transcriptional regulatory regions can be inherited, facilitating NF-κB binding and more robust subsequent activation. A high initial methylation level and CpG density around κB sites are correlated with the functional potential of transcriptional memory modules. Interestingly, the *CALCB* gene, encoding the proven migraine therapeutic target CGRP, exhibits the best transcriptional memory. A neighboring primate-specific endogenous retrovirus stimulates more rapid, more strong, and at least 100-fold more sensitive *CALCB* induction in subsequent TNF-α stimulation. Our study reveals that TNF-α-mediated transcriptional memory is governed by active DNA demethylation and greatly sensitizes memory genes to much lower doses of inflammatory cues.

*For correspondence:
zhubing@ibp.ac.cn

[†]These authors contributed equally to this work

Competing interests: The authors declare that no competing interests exist.

## Introduction

The epigenetic system has two important characteristic features: signal-induced plasticity and relatively stable mitotic inheritance. Signal-induced epigenetic plasticity allows changes in gene expression profiles and cell fate switching, whereas mitotic inheritance ensures the maintenance of gene expression profiles and cell fate. In theory, the combinatory effect of signal-induced plasticity and mitotic inheritance should convey epigenetic transcriptional memory, which allows certain genes to respond to previously experienced signals more robustly with faster kinetics and greater magnitude.

The first examples of transcriptional memory date back to almost half a century ago (*Bergink et al., 1973*). Current understanding of transcriptional memory events and underlying mechanisms has been nicely reviewed (*D'Urso and Brickner, 2014*). In *Saccharomyces cerevisiae*, the *INO1* and *GAL* genes are the best studied examples of transcriptional memory genes. These genes can remember past inositol or galactose induction and exhibit accelerated induction kinetics during a second induction, and this transcriptional memory can last for a few cell divisions. These transcriptional memory events depend on a number of mechanisms, including gene looping, nuclear pore targeting, histone H2A.Z deposition, and H3K4 methylation (*Bheda et al., 2020*; *D'Urso et al., 2016*; *Kundu et al., 2007*; *Light et al., 2010*; *Tan-Wong et al., 2009*). In mammals, interferon-induced transcriptional memory (*Gialitakis et al., 2010*; *Kamada et al., 2018*; *Light et al., 2013*) and inflammation-induced *Aim2* gene transcriptional memory (*Naik et al., 2017*) also rely on chromatin structure and histone modifications. On the other hand, DNA demethylation has been

**eLife digest** Genes are the instruction manuals of life and contain the information needed to build the building blocks that keep cells alive. To read these instructions, cells use specific signals that activate genes. The process, known as gene expression, is tightly controlled and for the most part, fairly stable. But gene expression can be modified in various ways.

Epigenetics is a broad term for describing reversible changes made to genes to switch them on and off. Sometimes, certain genes even develop a kind of 'transcriptional memory' where over time, their expression is enhanced and speeds up with repeated activation signals. But this may also have harmful effects.

For example, the signalling molecule called tumour necrosis factor α (TNF-α) is an essential part of the immune system. But it is also implicated in chronic inflammatory diseases, such as rheumatoid arthritis. In these conditions, cell signalling pathways triggering inflammation are overactive. One possibility is that TNF-α could be inducing the transcriptional memory of certain genes, amplifying their expression. But little is known about which fraction of genes exhibits transcriptional memory, and what differentiates memory genes from genes with stable expression.

Here, Zhao et al. treated cells grown in the laboratory with TNF-α to investigate its role in transcriptional memory and find out what epigenetic features might govern the process. The experiments showed that mimicking a sustained inflammation by stimulating TNF-α, triggered a transcriptional memory in some genes, and enabled them to respond to much lower levels of TNF-α on subsequent exposure.

Zhao et al. also discovered that genes tagged with methyl groups are more likely to show transcriptional memory when stimulated by TNF-α. However, they also found that these groups must be removed to consolidate any transcriptional memory.

This work shows how TNF-α influences can alter the expression of certain genes. It also suggests that transcriptional memory, stimulated by TNF-α, may be a possible mechanism underlying chronic inflammatory conditions. This could help future research in identifying more genes with transcriptional memory.

observed to associate with transcriptional memory of the *Tat* gene during glucocorticoid induction (*Thomassin et al., 2001*) and the *IL2* gene during T cell activation (*Murayama et al., 2006*), but whether DNA demethylation plays a causal role in transcriptional memory regulation remains unclear. In summary, transcriptional memory enhances the future response to a previously experienced stimulus, providing a mechanism for cells to adapt to environmental changes.

Several critical and intriguing questions regarding transcriptional memory need to be explored. Why do only a small fraction of genes induced by the same stimuli exhibit transcriptional memory? What differentiates genes with or without transcriptional memory effects? What are the characteristics of epigenetic modules governing transcriptional memory?

The proinflammatory cytokine TNF-α plays a vital role in the pathogenesis of chronic inflammatory diseases (*Reimold, 2003*; *Schett et al., 2013*). TNF-α activates the transcription factor NF-κB, which is essential for inflammatory responses (*Duh et al., 1989*; *Lowenthal et al., 1989*; *Osborn et al., 1989*; *Taniguchi and Karin, 2018*). Canonical NF-κB contains p65 and p50 and binds to genomic regions termed κB sites to activate target genes (*Hayden and Ghosh, 2008*; *Sen and Baltimore, 1986*; *Zhang et al., 2017*). We previously reported that short-term TNF-α treatment activates the methylated *IL32* gene in the absence of demethylation and that long-term TNF-α treatment induces DNA demethylation of the *IL32* promoter and CpG island, which depends on p65 and TET enzymes (*Zhao et al., 2019*). However, whether any NF-κB target gene exhibits transcriptional memory in response to TNF-α induction is unknown.

By combining reporter assays and genome-wide analysis, we found that the proinflammatory cytokine TNF-α can induce transcriptional memory effects in several target genes. These genes are associated with p65 peaks induced by TNF-α treatment. Interestingly, these regions are heavily methylated in cells naïve to TNF-α signaling and tend to become demethylated in a p65 and TET enzyme-dependent manner during sustained TNF-α treatment. Although p65 can associate with its binding motif located within heavily methylated regions, the transcriptional activation capacity of

p65 is potentiated by demethylation. This explains why a small fraction of p65 target genes display inflammatory transcriptional memory because NF-κB binding sites embedded within heavily methylated regions are more likely to serve as epigenetic memory modules. Our study reveals that the transcriptional memory established by sustained TNF-α stimulation is dependent on TET enzymes and significantly enhances the sensitivity to inflammatory signals.

## Results

### CMV promoter-driven reporter gene possesses inflammatory transcriptional memory

We previously generated HEK293F-derived cells containing an integrated *EGFP* reporter gene under the control of a fully methylated CMV (cytomegalovirus) promoter (*Figure 1A*), with which we performed a number of screens to identify factors and compounds regulating DNA methylation or methylation-mediated gene silencing (*Dong et al., 2018*; *Du et al., 2019*; *Li et al., 2018a*; *Li et al., 2018b*). Because the CMV promoter contains several κB sites (*Figure 1A*), we took advantage of the reporter cells and examined whether TNF-α treatment could activate the reporter gene and even induce transcriptional memory.

We first treated the reporter cells naïve to TNF-α with 50 ng/mL TNF-α (this concentration applies to all of the following experiments, unless indicated otherwise) for various lengths of time (*Figure 1B*), and the percentages of EGFP-positive cells were 0.96% (0 day), 43.7% (2 day), 56.3% (4 days), 69.0% (8 days), and 79.5% (12 days) (*Figure 1C*). Obviously, TNF-α treatment gradually potentiated the expression capacity of the reporter.

Next, we allowed the above cells to rest for 10 days in TNF-α-free media and then measured EGFP expression. Cells that had experienced longer TNF-α treatment tended to exhibit higher baseline expression (*Figure 1—figure supplement 1A*). Then, we applied a second TNF-α induction for 12 hr to test the transcriptional memory response (*Figure 1B*). Apparently, cells that had experienced longer initial TNF-α treatment exhibited much more robust EGFP expression, and EGFP-positive cells were 13.8% (0 day pretreatment), 18.6% (2 days pretreatment), 25.6% (4 days pretreatment), 40.1% (8 days pretreatment), and 56.5% (12 days pretreatment) of total cells (*Figure 1D*). These data indicate that not all initial TNF-α treatments were equal in inducing inflammatory transcriptional memory, and sufficient time was needed during the initial induction phase to consolidate transcriptional memory.

To directly compare the induction kinetics at the mRNA level, we performed RT-qPCR for the *EGFP* gene at various time points using cells naïve to TNF-α and cells that had experienced short-term (12 hr) or long-term (12 days) prior TNF-α treatment. Interestingly, we observed a clear transcriptional memory with faster and stronger induction only in cells that had experienced long-term prior TNF-α treatment (*Figure 1E*).

Together, the above experiments identified an example of inflammatory transcriptional memory and indicated that a consolidation phase during the initial treatment was required to establish such memory.

We then exposed naïve cells to TNF-α treatment for various lengths of time and measured the CMV promoter methylation levels, giving the following results: 90.6% (0 day), 86.1% (2 days), 76.7% (4 days), 56.7% (8 days), and 41.7% (12 days) (*Figure 1F*). Importantly, the CMV promoter methylation levels of cells experiencing different durations of TNF-α treatment were still maintained after 10 days of TNF-α withdrawal (*Figure 1—figure supplement 1B*).

The above results indicate that gradual CMV promoter demethylation induced by TNF-α treatment correlates with increasingly stronger EGFP expression and the consolidation of transcriptional memory. However, whether demethylation is required for memory consolidation needs to be determined.

### Inflammatory transcriptional memory of the CMV reporter depends on DNA demethylation mediated by TET enzymes

We noticed that in cells with 12 days of TNF-α treatment, the CMV promoter methylation level remained 41.7% (*Figure 1F*), and 20.5% of the cells remained EGFP-negative (*Figure 1C*). We suspected that not all cells underwent similar levels of demethylation even after 12 days. Therefore, we

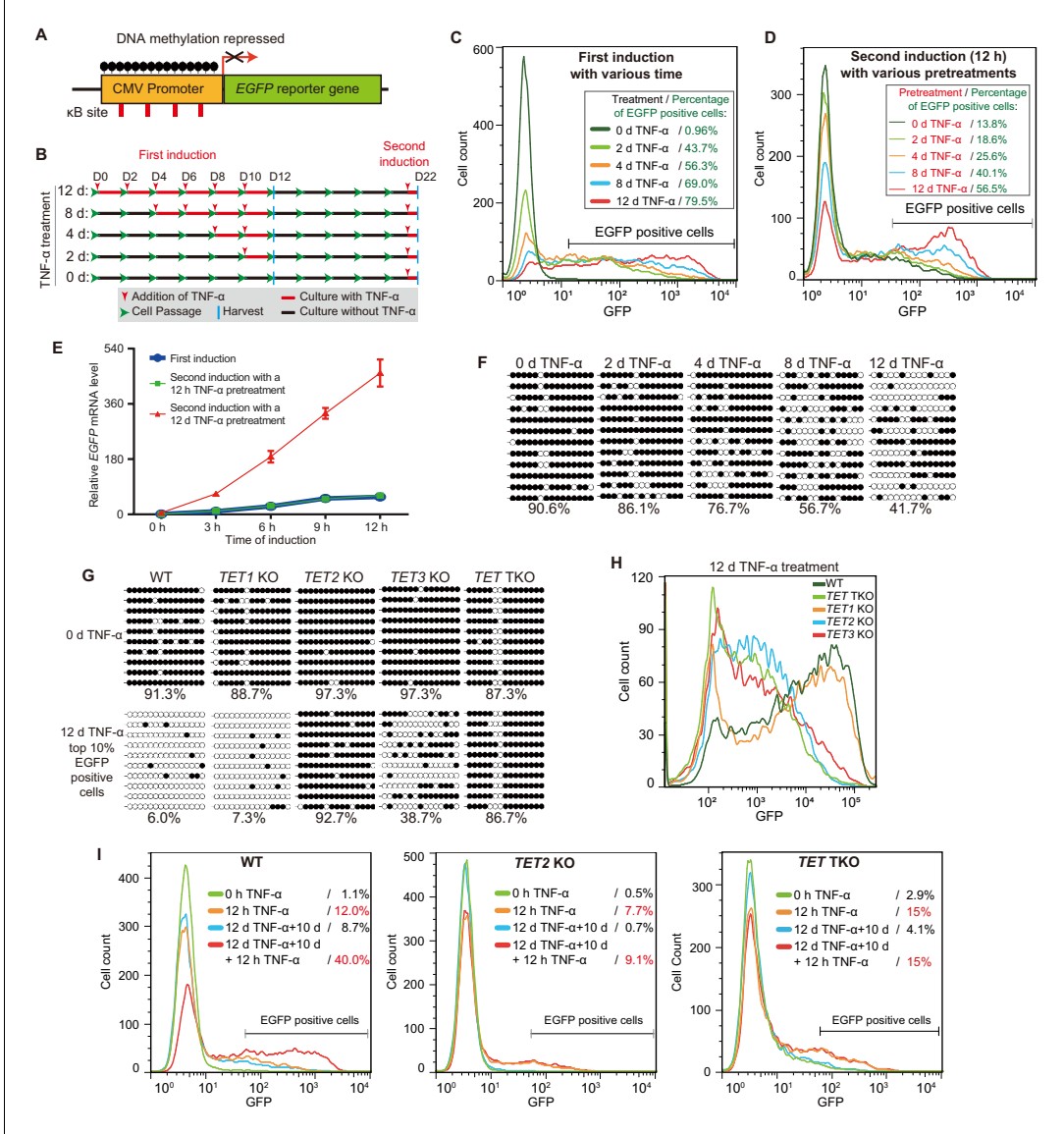

**Figure 1.** TET enzymes mediate the inflammatory transcriptional memory of a methylated CMV reporter. (**A**) Schematic of the *EGFP* reporter gene stably inserted into the HEK293F genome. The CMV promoter is highly modified and silenced by DNA methylation. (**B**) Experiment scheme. (**C**) Flow cytometry analysis of EGFP fluorescent intensity for the cells treated with TNF-α for 0 day, 2 days, 4 days, 8 days, and 12 days. (**D**) Flow cytometry analysis of EGFP fluorescent intensity for the cells with various pretreatments that received a 12-hr second TNF-α stimulation. (**E**) RT-qPCR results show the *EGFP* mRNA level in cells that received a first TNF-α induction and cells that received a second TNF-α induction after 12 hr or 12-day TNF-α treatment. The cells were cultured in TNF-α-free media for 10 days before receiving a second TNF-α induction. *GAPDH* is used as the internal control. Data are shown as mean ± SD from three independent experiments. Note: although not observable due to the scale of Y-axis, at 0 hr, *EGFP* mRNA level in cells with 12-day TNF-α pretreatment is fivefold higher than that in naïve cells. (**F**) Locus-specific bisulfite sequencing results of the CMV promoter for the cells treated with TNF-α for 0 day, 2 days, 4 days, 8 days, and 12 days. Filled circles, methylated CpGs; open circles, unmethylated CpGs. (**G**) The CMV promoter DNA methylation level of the top 10% of EGFP-positive cells sorted from 12-day-TNF-α-treated WT, *TET1* KO, *TET2* KO, *TET3* KO, and *TET* TKO cells by flow cytometry. (**H**) Flow cytometry analysis of EGFP fluorescent intensity for 12-day-TNF-α-treated WT, *TET1* KO, *TET2* KO, *TET3* KO, and *TET* TKO cells. (**I**) Flow cytometry analysis of the inflammatory transcriptional memory of the *EGFP* reporter in WT, *TET2* KO, and *TET* TKO cells.

The online version of this article includes the following source data and figure supplement(s) for figure 1:

**Source data 1.** Related to *Figure 1E*.

**Figure supplement 1.** TET enzymes mediate the inflammatory transcriptional memory of a methylated CMV reporter.

sorted the 10% of cells with the highest EGFP signal (*Figure 1—figure supplement 1C*) and measured their CMV promoter methylation level, which was 6.0% (*Figure 1G*), indicating that brighter cells had experienced more demethylation. To examine whether this is an active demethylation process, we treated *TET1* knockout (KO), *TET2* KO, *TET3* KO, and *TET* triple KO (*TET* TKO) cells (*Zhao et al., 2019*) with TNF-α for 12 days; sorted the 10% of cells with the highest EGFP signal; and measured their CMV promoter methylation levels. Obviously, TNF-α-induced CMV promoter demethylation mainly depended on TET2, with some contribution of TET3 and little participation of TET1 (*Figure 1G*).

Importantly, *TET2* KO, *TET3* KO, and *TET* TKO cells failed to reach the highest level of EGFP expression after 12 days of TNF-α treatment (*Figure 1H*), although their baseline expression (0 day) and expression after 1 day of TNF-α treatment were quite comparable to those of the wild-type cells (*Figure 1—figure supplement 1D*+E). Importantly, *TET2* KO and *TET* TKO cells also failed to exhibit a wild-type level of transcriptional memory during a second round of TNF-α treatment (*Figure 1I*, *Figure 1—figure supplement 1F*). These data demonstrate that TET enzyme–mediated DNA

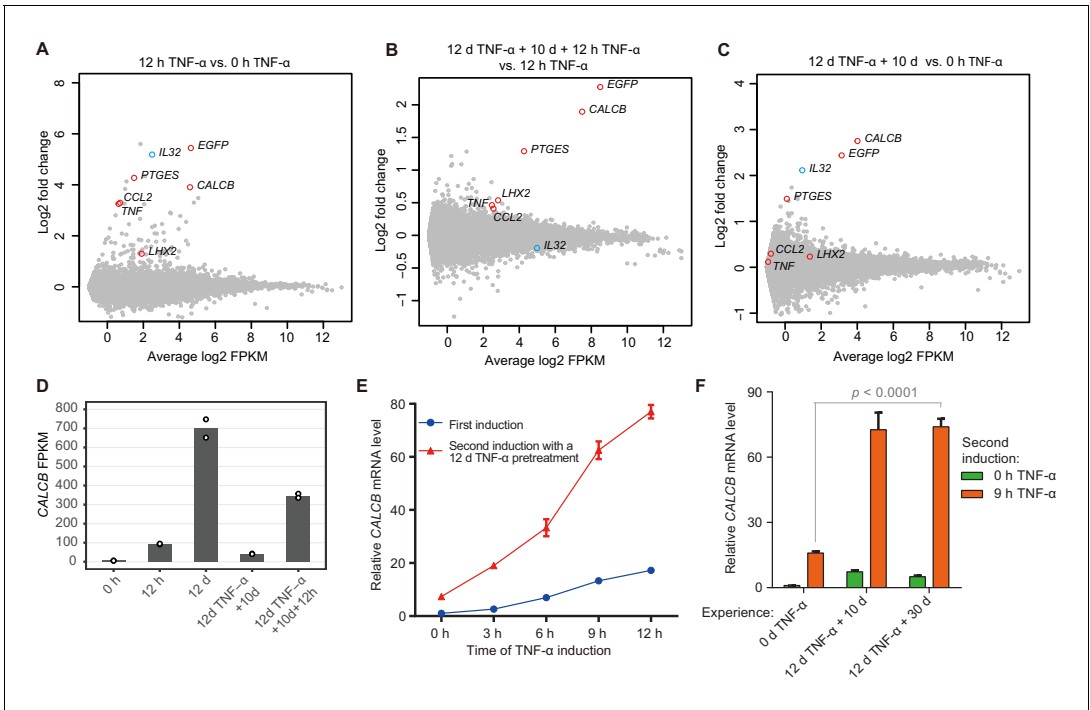

**Figure 2.** Identification of endogenous genes with inflammatory transcriptional memory. (**A**) Transcriptome changes between cells treated with TNF-α for 12 hr (12 hr TNF-α) and untreated cells (0 hr TNF-α). Red circles indicate genes with memory effect, and blue circle indicates *IL32*. (**B**) Transcriptome changes between cells received the second induction (12 days TNF-α + 10 days + 12 hr TNF-α-treated cells) and the first induction (12 hr TNF-α). (**C**) Transcriptome changes between cells rested for 10 days following a 12-day pretreatment (12 days TNF-α + 10 days) and untreated cells (0 hr TNF-α). (**D**) Expression levels (in FPKM) of *CALCB* in various treatment conditions. (**E**) RT-qPCR results show the *CALCB* mRNA level in cells that received a first TNF-α induction and in cells that received a second TNF-α induction after prior 12-day TNF-α treatment. The cells were cultured in TNF-α-free media for 10 days before receiving a second TNF-α stimulation. *GAPDH* is used as the internal control. Data are shown as mean ± SD from three independent experiments. (**F**) RT-qPCR results show the *CALCB* mRNA level. The cells that experienced 12 days of TNF-α treatment were cultured in TNF-α-free media for 10 days and 30 days, and then induced with a second TNF-α treatment for 9 hr. *GAPDH* is used as the internal control. Data are shown as mean ± SD from three independent experiments. One-tailed *t*-test.

The online version of this article includes the following source data and figure supplement(s) for figure 2:

**Source data 1.** Related to *Figure 2A*.
**Source data 2.** Related to *Figure 2B*.
**Source data 3.** Related to *Figure 2C*.
**Source data 4.** Related to *Figure 2D*.
**Source data 5.** Related to *Figure 2E*.
**Source data 6.** Related to *Figure 2F*.
**Figure supplement 1.** Identification of genes with transcriptional memory in response to TNF-α.

demethylation is required for the optimal activation and inflammatory transcriptional memory of the CMV reporter.

## *CALCB* and other endogenous genes display inflammatory transcriptional memory

From our experiences with the CMV reporter, we reasoned that the inflammatory memory genes likely possess the following features: 1. they should be activated by the inflammatory signal TNF-α; 2. they may display higher baseline expression in the absence of TNF-α signal after the initial induction; and 3. most importantly, they should respond to subsequent TNF-α treatment better. With the above features in mind, to identify endogenous genes with inflammatory transcriptional memory, we performed RNA-seq experiments with the following samples: no induction, 12 hr TNF-α induction, 12-day TNF-α induction, 10-day recovery from a 12-day TNF-α induction, and second induction (12 days of TNF-α treatment followed by 10 days without TNF-α, then treated with TNF-α for 12 hr).

We defined TNF-α-responsive genes as those that displayed greater than twofold upregulation upon 12 hr of TNF-α treatment with statistical significance (p<0.01) (*Figure 2A*). Among them, five endogenous genes together with the *EGFP* reporter were defined as inflammatory transcriptional memory genes, as they were expressed at least 1.3-fold higher in the second TNF-α induction than in the first TNF-α induction and exhibited an FPKM (fragments per kilobase of transcript per million mapped fragments) value greater than 5 (*Figure 2B*). The *EGFP* reporter and five endogenous memory genes can be categorized into two groups: *EGFP, CALCB,* and *PTGES* displayed excellent memory effects, whereas the memory effects of *CCL2, TNF,* and *LHX2* were relatively moderate (*Figure 2B*, *Figure 2—figure supplement 1A*). Indeed, *EGFP, CALCB,* and *PTGES* also displayed apparently elevated baseline expression in the absence of TNF-α signal after the initial induction (*Figure 2C*). *IL32* was upregulated in cells with 12-day TNF-α pretreatment (*Figure 2C*, *Figure 2—figure supplement 1A*; *Zhao et al., 2019*), but *IL32* did not exhibit a stronger second induction (*Figure 2B*, *Figure 2—figure supplement 1A*). Among the five endogenous genes displaying inflammatory transcriptional memory, *CALCB* was of particular interest for two reasons. First, *CALCB* clearly exhibited the best memory effect (*Figure 2B*, *Figure 2—figure supplement 1B*). Second, *CALCB* encodes β-CGRP. β-CGRP, and α-CGRP (encoded by *CALCA*) are two isoforms of CGRP (*c*alcitonin *g*ene-*r*elated *p*eptide). They are 37-amino acid–secreted neuropeptides and differ by three amino acids in humans (*Russell et al., 2014*; *Russo, 2015*). The two CGRP isoforms share similar biological activities and are potent vasodilators (*Brain et al., 1985*; *Russell et al., 2014*). The overexpression of CGRP is an important cause of migraine (*Edvinsson et al., 2018*; *Ho et al., 2010*; *Pellesi et al., 2017*; *Russell et al., 2014*; *Russo, 2015*; *Tepper, 2018*), and several monoclonal antibodies (eptinezumab, fremanezumab, galcanezumab, and erenumab) against CGRP or CGRP receptors have been approved by the FDA as therapeutics for migraine patients. For a gene whose elevated expression is key to its disease-causing effect (*Ashina et al., 2000*; *Fusayasu et al., 2007*; *Goadsby et al., 1990*), it is certainly important to know under which circumstances this gene can reach a very high level of expression. We noticed that *CALCB* gene expression reached very high levels in some conditions we analyzed, with an average FPKM value of 699 after 12 days of TNF-α treatment and 346 after 12 hr of TNF-α treatment in cells with 12 days of prior TNF-α exposure (*Figure 2D*).

We then measured *CALCB* induction kinetics in cells with or without prior TNF-α exposure, and clearly, *CALCB* exhibited robust induction with faster kinetics and stronger expression during subsequent induction (*Figure 2E*). Importantly, the transcriptional memory effect of *CALCB* was maintained for at least 30 days in the absence of TNF-α (*Figure 2F*). Therefore, we selected *CALCB* as a paradigm to further study the mechanisms governing inflammatory transcriptional memory.

Importantly, the transcriptomes of the naïve cells and memory consolidated cells were extremely similar ($r = 0.999$), and the same was true for the transcriptomes of the first and second inductions ($r = 0.999$) (*Figure 2—figure supplement 1C*), indicating that the general physiology of the cells was not altered during memory consolidation, except for genes primed for memory response.

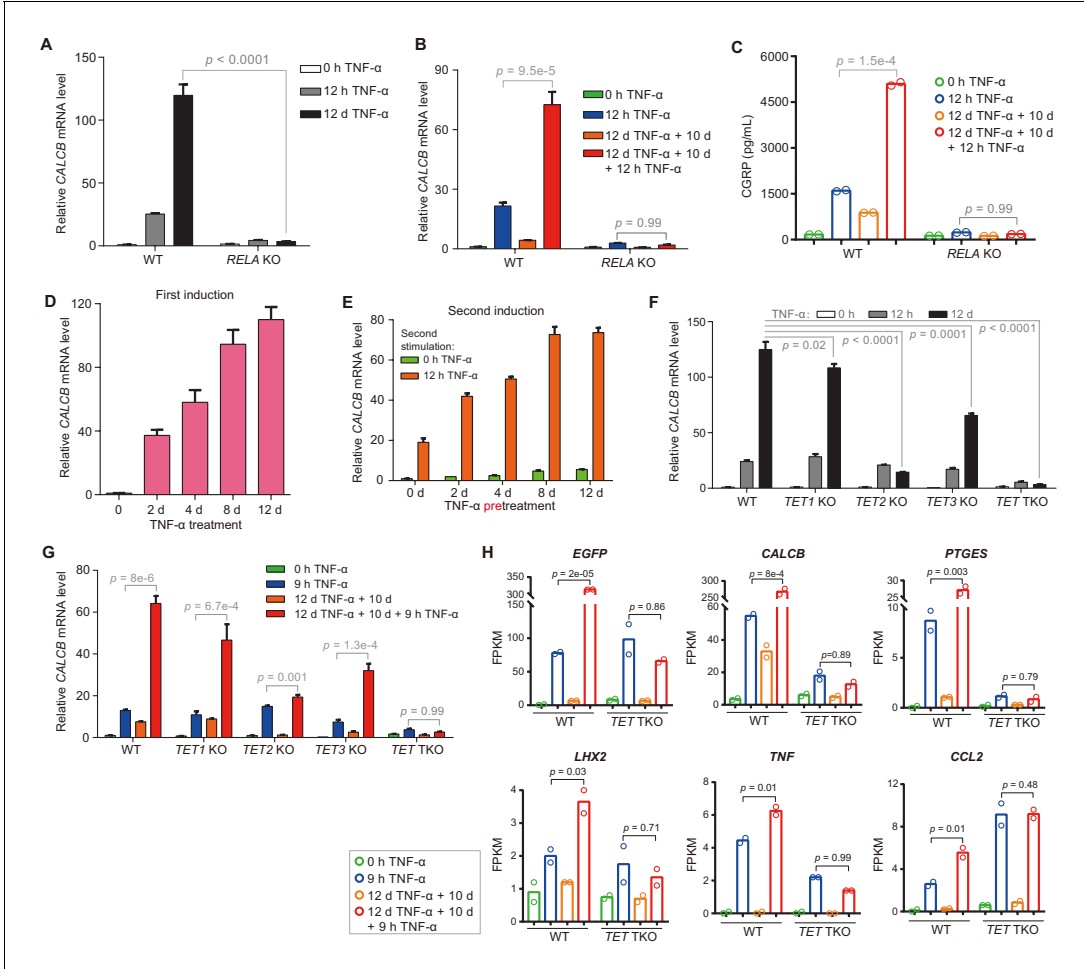

**Figure 3.** The inflammatory transcriptional memory of *CALCB* is dependent on p65 and TET enzymes. (**A**) RT-qPCR results show the *CALCB* mRNA levels at 0 hr, 12 hr, and 12 days in TNF-α-treated WT and *RELA* KO cells. *GAPDH* is used as the internal control. Data are shown as mean ± SD from three independent experiments. Two-tailed *t*-test. (**B**) RT-qPCR results show the transcriptional memory of *CALCB* in WT and *RELA* KO cells in response to TNF-α. *GAPDH* is used as the internal control. Data are shown as mean ± SD from three independent experiments. One-tailed *t*-test. (**C**) Sandwich ELISA results show the CGRP release level in the media for WT and *RELA* KO cells in response to TNF-α. Data are shown as the mean from two independent experiments. One-tailed *t*-test. (**D**) RT-qPCR results show the *CALCB* mRNA levels in WT cells treated with TNF-α for 0 day, 2 days, 4 days, 8 days, and 12 days. *GAPDH* is used as the internal control. Data are shown as mean ± SD from three independent experiments. (**E**) RT-qPCR results show the *CALCB* mRNA level in WT cells with various pretreatments that received a second TNF-α induction for 12 hr. *GAPDH* is used as the internal control. Data are shown as mean ± SD from three independent experiments. (**F**) RT-qPCR results show the *CALCB* mRNA level in WT, *TET1* KO, *TET2* KO, *TET3* KO, and *TET* TKO cells treated with TNF-α stimulation for 0 hr, 12 hr, and 12 days. *GAPDH* is used as the internal control. Data are shown as mean ± SD from three independent experiments. Two-tailed *t*-test. (**G**) RT-qPCR results show the transcriptional memory of *CALCB* in WT, *TET1* KO, *TET2* KO, *TET3* KO, and *TET* TKO cells in response to TNF-α. *GAPDH* is used as the internal control. Data are shown as mean ± SD from three independent experiments. One-tailed *t*-test. (**H**) Expression levels (in FPKM) of *EGFP*, *CALCB*, *PTGES*, *LHX2*, *TNF*, *CCL2* for WT, and *TET* TKO cell in various treatment conditions. Data are shown as the mean from two independent experiments. One-tailed *t*-test.

The online version of this article includes the following source data and figure supplement(s) for figure 3:

**Source data 1.** Related to *Figure 3A*.
**Source data 2.** Related to *Figure 3B*.
**Source data 3.** Related to *Figure 3C*.
**Source data 4.** Related to *Figure 3D*.
**Source data 5.** Related to *Figure 3E*.
**Source data 6.** Related to *Figure 3F*.
**Source data 7.** Related to *Figure 3G*.
**Source data 8.** Related to *Figure 3H*.
**Figure supplement 1.** The inflammatory transcriptional memory of *CALCB* and *PTGES*.

## *CALCB* expression requires p65, and its inflammatory transcriptional memory requires TET enzymes

The deletion of the *RELA* gene (*Zhao et al., 2019*), which encodes p65, severely impaired TNF-α-induced *CALCB* activation (*Figure 3A*) and the inflammatory transcriptional memory of *CALCB* (*Figure 3B*). Since *CALCA* is barely transcribed (*Figure 3—figure supplement 1A*), we were able to measure β-CGRP secreted in the media by performing sandwich ELISA using antibodies against CGRP. Likewise, the induction and memory effect of secreted β-CGRP at the protein level was abrogated in *RELA* KO cells (*Figure 3C*).

On the other hand, *CALCB* expression increased gradually during induction (*Figure 3D*), and cells that had experienced longer initial TNF-α exposure displayed higher expression with subsequent induction (*Figure 3E*). These features were highly similar to the behavior of the methylated CMV reporter (*Figure 1*) and prompted us to ask whether *CALCB* transcriptional memory is also mediated by DNA demethylation. Indeed, *TET2* KO and *TET* TKO cells displayed great defects in reaching the induction peak after 12 days of TNF-α treatment (*Figure 3F*) and failed to maintain robust transcriptional memory during the second round of TNF-α treatment (*Figure 3G*). These results clearly indicate that DNA demethylation mediated by TET enzymes is critical for the inflammatory transcriptional memory of *CALCB*.

Another gene with excellent inflammatory transcriptional memory is *PTGES* (*Figure 2B*), which encodes prostaglandin E synthase that synthesizes a key inflammatory mediator prostaglandin $E_2$ (*Gomez et al., 2013*; *Jakobsson et al., 1999*). Similar to *CALCB*, *PTGES* also reached higher expression during long-term TNF-α induction (*Figure 3—figure supplement 1B*), and *RELA* KO disrupted its TNF-α-induced activation and transcriptional memory (*Figure 3—figure supplement 1C*).

To confirm the roles of TET enzymes in regulating other inflammatory transcriptional memory genes, we performed RNA-seq experiments with *TET* TKO cells under four conditions: no induction, 9 hr of TNF-α induction, 10 days of recovery from a 12-day TNF-α induction, and second induction (12 days of TNF-α treatment followed by 10 days without TNF-α, then induced with TNF-α for 9 hr) and compared the results with wild-type cells. The elevated expression of all inflammatory transcriptional memory genes during the second TNF-α treatment was abrogated in *TET* TKO cells (*Figure 3H*). Therefore, we conclude that inflammatory transcriptional memory is controlled by DNA demethylation mediated by TET enzymes.

## Identification of putative TNF-α-responsive elements for *CALCB* and other inflammatory transcriptional memory genes

The *CALCB* promoter was unmethylated even in untreated cells (*Figure 3—figure supplement 1D*). This prompted us to ask whether *CALCB* has a distal TNF-α-responsive element that governs its inflammatory transcriptional memory.

To identify putative TNF-α-responsive elements, we performed ChIP-seq experiments to profile p65 and H3K27ac occupancy in the following samples: no induction, 12 hr of TNF-α induction, 10-day recovery from a 12-day TNF-α induction, and second induction (12 days of initial TNF-α induction followed by 10 days of recovery and then a second TNF-α induction for 12 hr).

We identified 405 TNF-α-induced p65 peaks that were co-occupied by H3K27ac (*Figure 4A*), the best known chromatin indicator of active transcriptional regulatory elements (*Creyghton et al., 2010*; *Heintzman et al., 2009*; *Zhang et al., 2020*), and we defined them as TNF-α-responsive elements. The five endogenous inflammatory transcriptional memory genes were associated with 10 TNF-α-responsive elements (*Figure 4B*, *Figure 4—figure supplement 1A*). For *CALCB*, two TNF-α-responsive elements were observed at the upstream distal regions (−10 kb and −3 kb) of *CALCB*, which we considered putative enhancers of *CALCB*, and they displayed TNF-α-induced p65 and H3K27ac occupancy (*Figure 4B*). Most importantly, p65 and H3K27ac exhibited further elevation of TNF-α-responsive elements associated with *CALCB* and *PTGES* (*Figure 4B*, *Figure 4—figure supplement 1A*), which were the two endogenous genes displaying the best inflammatory transcriptional memory (*Figure 2B*).

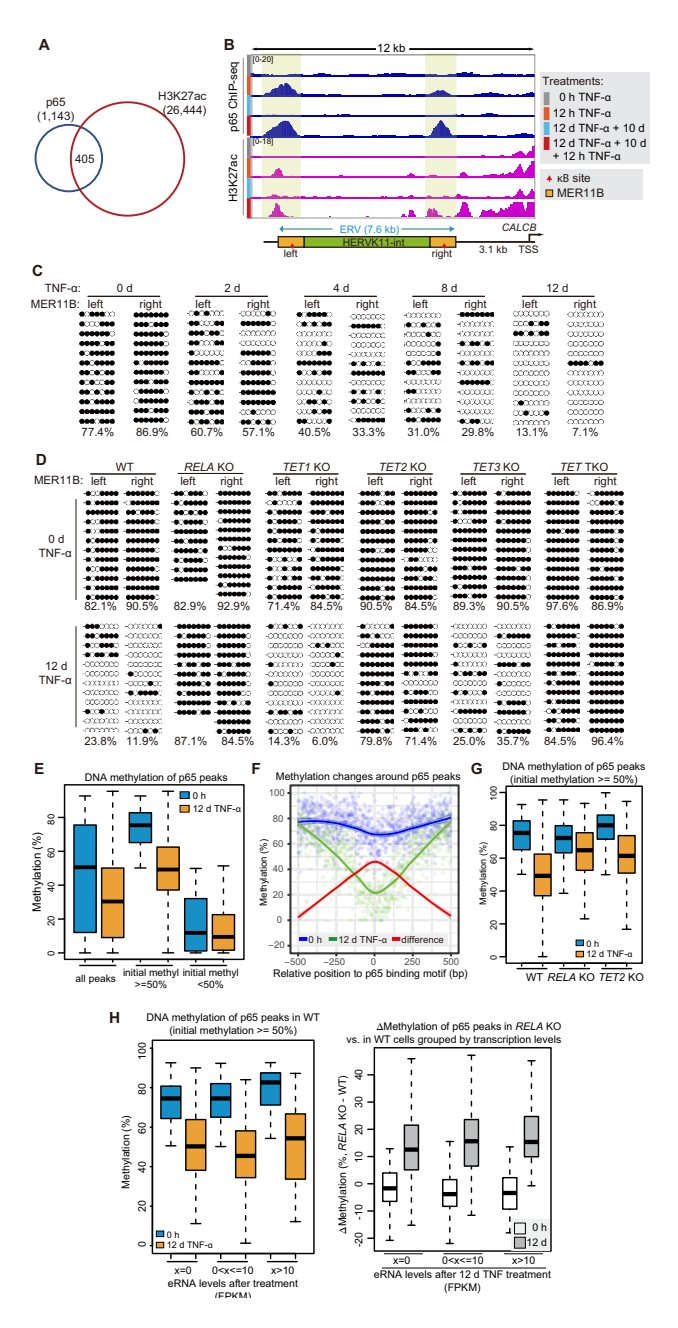

**Figure 4.** DNA demethylation of TNF-α-responsive elements during memory consolidation depends on TET enzymes and p65. (**A**) Overlap between p65 ChIP-seq peaks and H3K27ac ChIP-seq peaks in cells treated with TNF-α for 12 hr. (**B**) Genome browser view shows p65 occupancy and H3K27ac ChIP-seq signal at the endogenous retrovirus (ERV) region upstream of *CALCB* gene. (**C**) Locus-specific bisulfite sequencing results of MER11B-left LTR and MER11B-right elements for the cells treated with TNF-α for 0 day, 2 days, 4 days, 8 days, and 12 days. (**D**) Locus-specific bisulfite sequencing results show the DNA methylation level of MER11B-left LTR and MER11B-right elements for 0-day and 12-day TNF-α-treated WT, *TET1* KO, *TET2* KO, *TET3* KO, *TET* TKO, and *RELA* KO cells. (**E**) Changes of DNA methylation of p65 peaks in 12-day TNF-α-treated cells vs. 0 hr treated cell. Highly methylated and lowly methylated regions are also shown separately. (**F**) DNA methylation around p65 peaks. Colored dots indicate each CpG by relative positions to the binding motif with the highest score. Lowess-smoothed curves were drawn for 0 hr (blue), 12-day TNF-α treatment (green), and the differences between them (red). (**G**) Average methylation level of p65 peaks in WT, *RELA* KO, and *TET2* KO cell with 0 hr or 12-day TNF-α treatment. Only p65 peaks that were initially methylated (methylation level ≥50%) are plotted. (**H**) Left panel shows average

*Figure 4 continued on next page*

*Figure 4 continued*

methylation level of p65 peaks in WT cells with 0 hr or 12 days TNF-α treatment, categorized by three indicated transcription levels (eRNA at peaks normalized in RPKM). Right panel shows the methylation difference of p65 peaks in *RELA* KO in comparison to that in WT cells, with or without 12-day TNF-α treatment. Similar to panel G, p65 peaks that were initially methylated (methylation level ≥50%) are used in panel H.

The online version of this article includes the following source data and figure supplement(s) for figure 4:

**Source data 1.** Related to *Figure 4E and G*.
**Source data 2.** Related to *Figure 4F*.
**Figure supplement 1.** The DNA methylation of MER11B elements adjacent to *CALCB* can be maintained after TNF-α withdrawal.

## DNA demethylation occurs at putative *CALCB* distal enhancers residing in an endogenous retrovirus during memory consolidation

Interestingly, the two TNF-α-induced enhancers associated with the *CALCB* gene resided in a 7 kb endogenous retrovirus (ERV) spanning from −10 kb to −3 kb upstream of the transcription start site of *CALCB* (*Figure 4B*). Since these two enhancers are located at the left and right end of an ERV, and each enhancer corresponds to a MER11B element, we termed them MER11B-left and MER11B-right.

Both MER11B elements contain a κB site flanked by 7 CpG sites within regions spanning approximately 200 bp (209 bp for MER11B-left and 166 bp for MER11B-right; *Figure 4—figure supplement 1B*). We measured the DNA methylation levels of these two regions, and both regions were highly methylated in cells naïve to TNF-α signaling and became gradually demethylated during TNF-α treatment (*Figure 4C*). The demethylated states were well maintained after culturing for 10 days in the absence of TNF-α (*Figure 4—figure supplement 1C*) and could even last for 30 days (*Figure 4—figure supplement 1D*), indicating that long-term TNF-α exposure, in certain cases, may permanently alter the future signal response. Similar to CMV promoter demethylation, TNF-α-induced *CALCB* enhancer demethylation depended on p65 and the TET enzymes, with a major contribution from TET2 (*Figure 4D*).

## DNA demethylation occurs at p65 peaks in a binding-dependent, transcriptional activity-independent manner

To monitor DNA demethylation at p65 peaks associated with inflammatory transcriptional memory and all other p65 peaks, we performed enzymatic methyl-seq for genome-wide DNA methylation analysis with wild-type, *RELA* KO and *TET* TKO cells under no induction or 12 days of TNF-α induction. Indeed, DNA demethylation occurred at p65 peaks (*Figure 4E*), and the demethylation effect was stronger in regions adjacent to the peak centers (*Figure 4F*). The deletion of *RELA* impaired the above demethylation events, and the deletion of *TET2* reduced the extent of demethylation (*Figure 4G*).

To differentiate the contributions of p65 binding and p65-dependent transcriptional activation in TNF-α-induced demethylation, we determined p65 binding peaks in TNF-α-treated cells, selected those that were heavily methylated in naïve cells, and then categorized them into three groups based on their eRNA expression levels. Interestingly, regardless of whether eRNA could be detected from these p65 binding peaks, demethylation occurred in a similar way (*Figure 4H*). Therefore, we conclude that p65 binding per se, but not active transcription, is the main cause of TNF-α-induced demethylation, which is in line with an early observation that DNA-binding proteins can induce DNA demethylation without gene activation (*Stadler et al., 2011*).

## ERV containing *CALCB* enhancers is required for TNF-α-induced *CALCB* expression and transcriptional memory

We identified putative TNF-α-responsive enhancers for *CALCB* based on the occupancy of p65 and H3K27ac and established a correlation between NF-κB-induced demethylation at these enhancers and the transcriptional memory of *CALCB*. To provide direct evidence for the functional significance of these enhancers, we generated KO cells by deleting MER11B-left (*Figure 5—figure supplement 1A*), MER11B-right (*Figure 5—figure supplement 1B*), both MER11B elements (*Figure 5—figure*

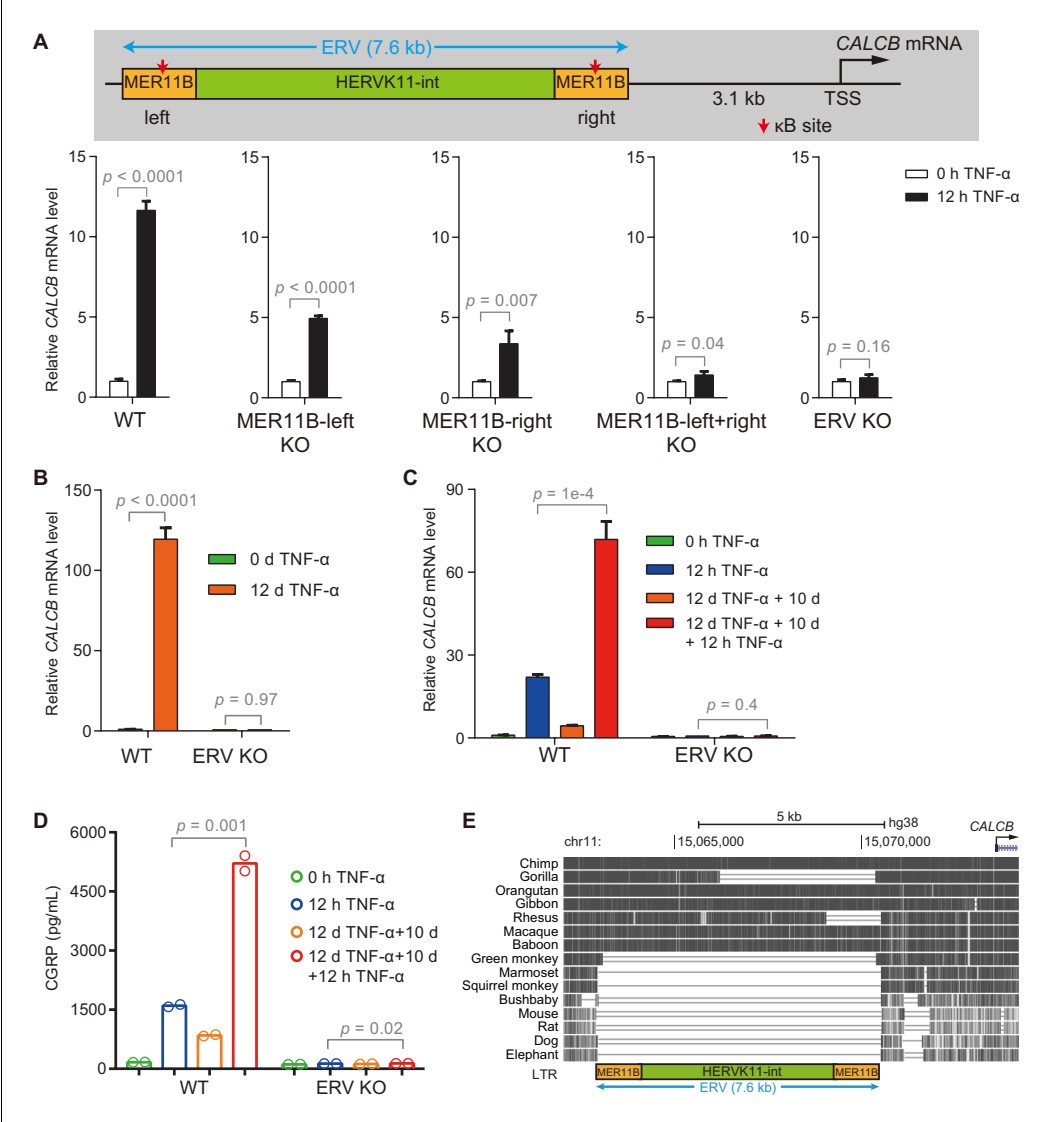

**Figure 5.** The ERV upstream of *CALCB* is required for its TNF-α-induced activation and transcriptional memory. (**A**) RT-qPCR results show the *CALCB* mRNA level of WT, MER11B-left KO, MER11B-right KO, MER11B-left+right KO, and ERV KO cells treated with TNF-α for 0 hr and 12 hr. *GAPDH* is used as the internal control. Data are shown as mean ± SD from three independent experiments. Two-tailed *t*-test. (**B**) RT-qPCR results show *CALCB* transcriptional levels at 0 hr and 12 days in TNF-α-treated WT and ERV KO cells. *GAPDH* is used as the internal control. Data are shown as mean ± SD from three independent experiments. Two-tailed *t*-test. (**C**) RT-qPCR results show the transcriptional memory of *CALCB* in WT and ERV KO cells in response to TNF-α. *GAPDH* is used as the internal control. Data are shown as mean ± SD from three independent experiments. One-tailed *t*-test. (**D**) Sandwich ELISA results show the CGRP release level in the media for WT and ERV KO cells in response to TNF-α. Data are shown as the mean from two independent experiments. One-tailed *t*-test. (**E**) UCSC genome browser track shows multiple alignments of the ERV region upstream of the *CALCB* gene in selected vertebrate species.

The online version of this article includes the following source data and figure supplement(s) for figure 5:

**Source data 1.** Related to *Figure 5B*.
**Source data 2.** Related to *Figure 5C*.
**Source data 3.** Related to *Figure 5D*.
**Figure supplement 1.** Validation of MER11B-left KO cells and MER11B-right KO cells.
**Figure supplement 2.** Validation of MER11B-left+right KO cells and ERV KO cells.

supplement 2A), and the entire ERV (*Figure 5—figure supplement 2B*). The activation of *CALCB* was partially impaired in MER11B-left and MER11B-right KO cells (*Figure 5A*), suggesting that both MER11B elements were involved in *CALCB* activation mediated by TNF-α. On the other hand, cells

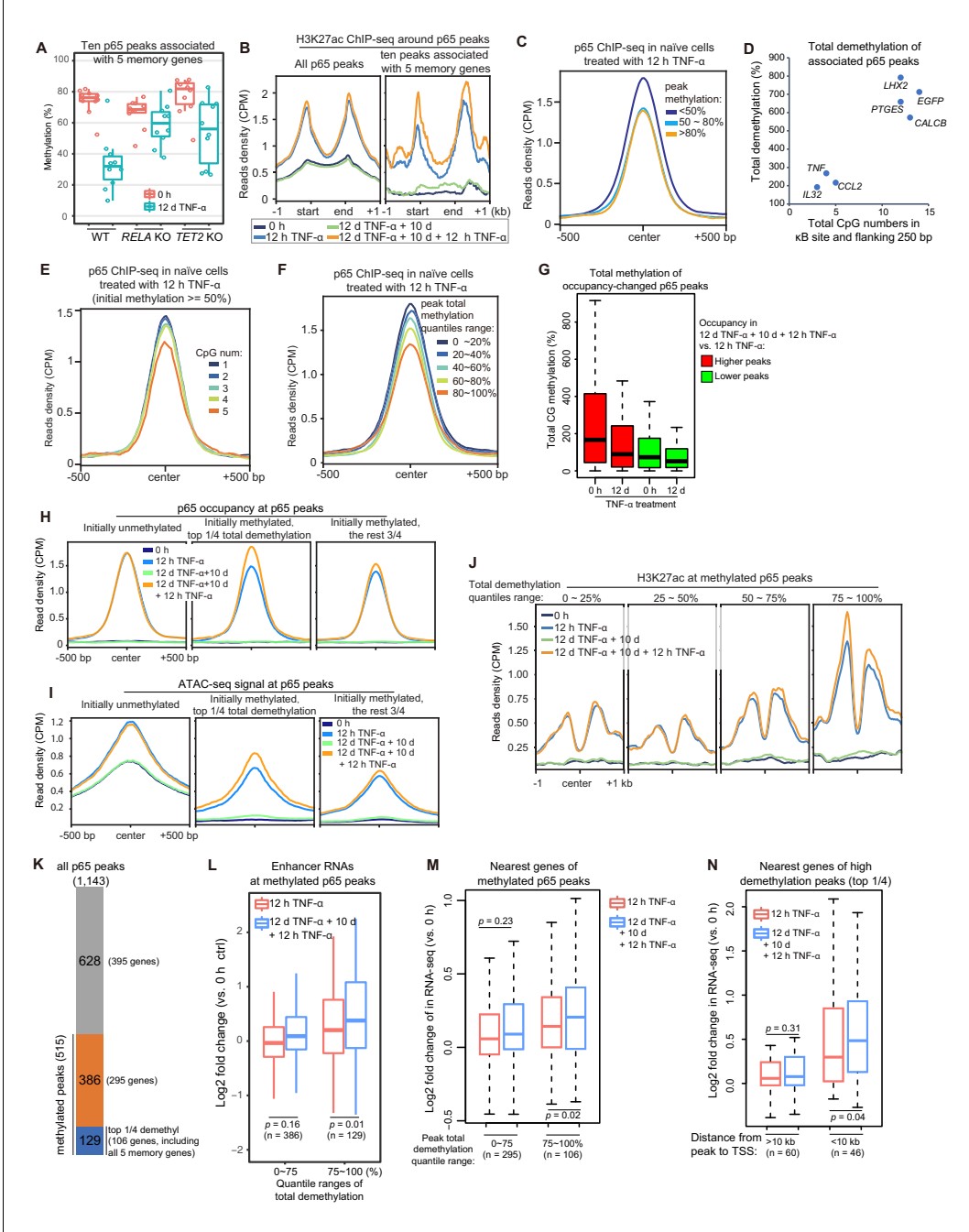

**Figure 6.** High initial methylation level and CpG density determine the functional potential of transcriptional memory modules in response to TNF-α stimulation. (**A**) Methylation levels of ten p65 peaks that are associated with the five memory genes. Average values of p65 peaks are shown as circles for 0 hr and 12 days TNF-α-treated WT, *RELA* KO and *TET2* KO cells, respectively. (**B**) H3K27ac ChIP-seq profiles around all p65 peaks (left) and ten peaks that are associated with five memory genes (right). Different colors indicate cells with various treatments. (**C**) Averaged profile of p65 ChIP-seq at p65 peaks grouped by various peak methylation levels. In addition to sequencing depth, read counts located at initially unmethylated p65 peaks (methylation level less than 20%) were used to determine the scale factor for normalization of ChIP-seq signals. Peak-centered 1 kb region are shown. (**D**) Five memory genes, together with *EGFP* and *IL32*, are plotted for their CpG numbers and total demethylation of associated p65 peaks in the flanking 250 bp regions of κB sites. (**E**) Averaged profile of p65 ChIP-seq at methylated p65 peaks grouped by CpG numbers. (**F**) Averaged profile of p65 ChIP-seq at p65 peaks grouped by total methylations in five ranges of quantiles. (**G**) Total methylation level of occupancy-changed p65 peaks that were treated with 0 hr or 12 days TNF-α. Red columns stand for p65 peaks that are relatively higher in the second

*Figure 6 continued on next page*

*Figure 6 continued*

induction, and green ones stand for p65 peaks that are relatively lower in the second induction (as indicated in *Figure 6—figure supplement 1*). (H) Averaged p65 occupancy at p65 peaks that are categorized into three groups by initial methylation level and total demethylations in 12-day TNF-α treatment vs. 0 hr. The left plot shows initially unmethylated p65 peaks (mCG <20%), the middle one shows initially methylated p65 peaks with the top 25% total demethylation after 12-day TNF-α treatment (higher than the upper quartile, Q3), and the right one shows initially methylated p65 peaks with the lowest 75% total demethylation. The four treatments are shown in indicated colors. (I) Averaged ATAC-seq profile at p65 peaks that are categorized into three groups by the same criteria as that used in panel H. (J) Averaged profile of H3K27ac ChIP-seq at methylated p65 peaks that are grouped into four ranges of quantiles by total demethylations in 12-day TNF-α treatment vs. 0 hr. The four treatments are shown in indicated colors. (K) The number of p65 peaks and their associated genes in designated groups. (L) Expression changes of enhancer RNAs at p65 peaks with different degree of demethylation. Paired two-tailed *t*-test. (M) Expression changes of nearest genes of methylated p65 peaks. Paired two-tailed *t*-test. (N) Impact of distance between p65 peaks and TSSs of neighboring genes. Paired two-tailed *t*-test.

The online version of this article includes the following source data and figure supplement(s) for figure 6:

**Source data 1.** Related to *Figure 6A*.
**Source data 2.** Related to *Figure 6D*.
**Source data 3.** Related to *Figure 6L*.
**Source data 4.** Related to *Figure 6M*.
**Source data 5.** Related to *Figure 6N*.
**Figure supplement 1.** MA-plot of p65 occupancy in two times of induction.

lacking both MER11B elements or the entire ERV completely failed to respond to TNF-α signaling (*Figure 5A and B*). Cells lacking the entire ERV did not display any inflammatory transcriptional memory of *CALCB* (*Figure 5C*) or secreted β-CGRP (*Figure 5D*). Notably, although the *CALCB* gene is conserved between rodents and humans, this regulatory ERV only exists in some primates (*Figure 5E*), suggesting that the inflammatory transcriptional memory of *CALCB* emerged during evolution after an ERV was inserted in the neighborhood and that this ERV was coopted as the epigenetic memory module.

## High initial methylation level and CpG density around the κB sites determine the functional potential of inflammatory transcriptional memory modules

Although we identified only five endogenous inflammatory transcriptional memory genes, it is logical to hypothesize that what we discovered represents a general mechanism and that other genes may exhibit inflammatory transcriptional memory in other cell types and/or signal contexts. Therefore, it is highly important to determine the characteristic features of the inflammatory transcriptional memory modules associated with these five genes. The p65 peaks were linked to the nearest genes that are located within a 100 kb distance, and in total, 10 TNF-α-induced p65 peaks were associated with these genes. These peaks displayed several consistent features: they were highly methylated in cells naïve to TNF-α and exhibited obvious demethylation during memory consolidation, their demethylation was impaired upon the loss of *RELA* and *TET2* (*Figure 6A*), and they exhibited greater H3K27ac enrichment during the second TNF-α induction (*Figure 6B*). These results imply that although TNF-α-induced p65 chromatin association can occur without DNA demethylation, the p65 chromatin association is favored in regions with less methylation. Indeed, when we grouped p65 peaks according to their initial DNA methylation levels and then plotted p65 occupancy in cells treated with the first dose of TNF-α for 12 hr, it was obvious that p65 could bind methylated regions but favored unmethylated regions (*Figure 6C*). Thus, one important mechanism in consolidating inflammatory transcriptional memory is to change initially disfavored methylated p65-binding regions into favored unmethylated regions, to facilitate p65 binding and to achieve more effective transcriptional activation when a subsequent stimulus arrives.

Active DNA demethylation mediated by the TET enzymes was required for consolidating transcriptional memory (*Figure 3H*), but not all genes with TNF-α-induced demethylation at their regulatory regions exhibited transcriptional memory. For example, the *IL32* promoter and CpG islands

were demethylated after 12 days of TNF-α treatment (*Zhao et al., 2019*), but *IL32* did not exhibit inflammatory transcriptional memory (*Figure 2B*). Why?

To address this critical question, we carefully compared p65 peaks in the *IL32* promoter and other transcriptional memory genes, and we noticed that the adjacent region of the κB site in the *IL32* promoter had fewer CpGs than the other p65 peaks associated with genes displaying a memory effect (*Figure 6D*), whereas the three genes (*EGFP, CALCB,* and *PTGES*) with the best memory effects (*Figure 2B*) contained more CpGs in their associated p65 peaks (*Figure 6D*). To test the effect of the number of CpGs, we chose highly methylated p65 peaks (methylation level ≥50%), grouped them according to the CpG number within 250 bp flanking the κB sites, and then plotted p65 occupancy in cells treated with the first dose of TNF-α for 12 hr. Indeed, p65 occupancy was higher in regions with fewer CpGs even though the methylation levels were equally high (*Figure 6E*).

The above results suggest that the total DNA methylation (combining CpG number and methylation level) in the adjacent regions of κB sites is critical for their potential to serve as epigenetic modules for inflammatory transcriptional memory. Indeed, when we grouped p65 peaks according to their total methylation level and plotted p65 occupancy in cells treated with the first dose of TNF-α for 12 hr, regions with higher total methylation were clearly more disfavored for p65 binding (*Figure 6F*). In retrospect, this is easy to understand because the efficiency of methylation-mediated repression depends not only on the methylation level but also on the regional methylated CpG density (*Curradi et al., 2002*; *Hsieh, 1994*).

We systematically analyzed the occupancy of p65 between the two TNF-α induction times (*Figure 6—figure supplement 1*) and observed that occupancy-increased peaks were prone to high initial methylation and became demethylated after 12 days of TNF-α treatment (*Figure 6G*). We also compared p65 occupancy, ATAC-seq signal, and H3K27ac signal according to various levels of total methylated CpGs, and the reduction in the number of methylated CpGs promoted p65 binding, chromatin opening, and H3K27ac (*Figure 6H–J*).

The above results suggest that initially methylated p65 peaks with the largest amount of total demethylation may function as potential inflammatory transcriptional memory modules. To functionally assess this hypothesis, we chose 515 methylated p65 peaks (methylation level ≥50%) out of 1143 p65 peaks and divided them into two groups: the highly demethylated p65 peaks comprising the top 25% of total demethylation during 12-day TNF-α treatment (129 peaks associated with 106 genes, including all five memory genes) and the other methylated p65 peaks (386 peaks associated with 295 genes; *Figure 6K*). Then, we analyzed these peaks and their associated genes.

First, upon second induction, the genes of the highly demethylated group displayed higher expression of their own enhancer RNAs (*Figure 6L*). This suggests that these p65 peak regions possess the potential ability to serve as inflammatory epigenetic memory modules. Moreover, neighboring genes associated with the highly demethylated p65 peaks also exhibited higher expression during the second induction (*Figure 6M*). In addition, the impact on neighboring genes was distance-sensitive, and genes with transcriptional start sites (TSSs) within 10 kb of the highly demethylated p65 peaks exhibited stronger transcriptional memory (*Figure 6N*).

Taken together, p65 peaks with high initial methylation and CpG density are more likely to serve as inflammatory transcriptional memory modules.

## Memory consolidation elevates TNF-α sensitivity by more than 100-fold

Transcriptional memory has been shown to allow cells to respond to previously exposed stimuli more rapidly and more strongly (*Bergink et al., 1973*; *D'Urso and Brickner, 2014*; *D'Urso et al., 2016*; *Gialitakis et al., 2010*; *Kamada et al., 2018*; *Light et al., 2010*; *Light et al., 2013*; *Murayama et al., 2006*; *Naik et al., 2017*; *Tan-Wong et al., 2009*; *Thomassin et al., 2001*). In theory, transcriptional memory may also allow cells to respond to previously exposed stimuli in a much more sensitive manner, which has rarely been studied. We have already demonstrated that *CALCB* responded to subsequent TNF-α stimuli with faster kinetics and greater magnitude (*Figure 2E*), but we have not tested whether *CALCB* can respond to a much weaker TNF-α signal after memory consolidation. Given the substantial demethylation at MER11B-left and MER11B-right after memory consolidation (*Figure 4C*), which removes the epigenetic barrier for transcriptional activation and promotes p65 association (*Figure 6C*), we speculated that MER11B-left and MER11B-right demethylation might allow the *CALCB* gene to respond to TNF-α signaling in a much more sensitive way. Therefore, we treated naïve cells and cells that had experienced a 12-day initial TNF-α treatment

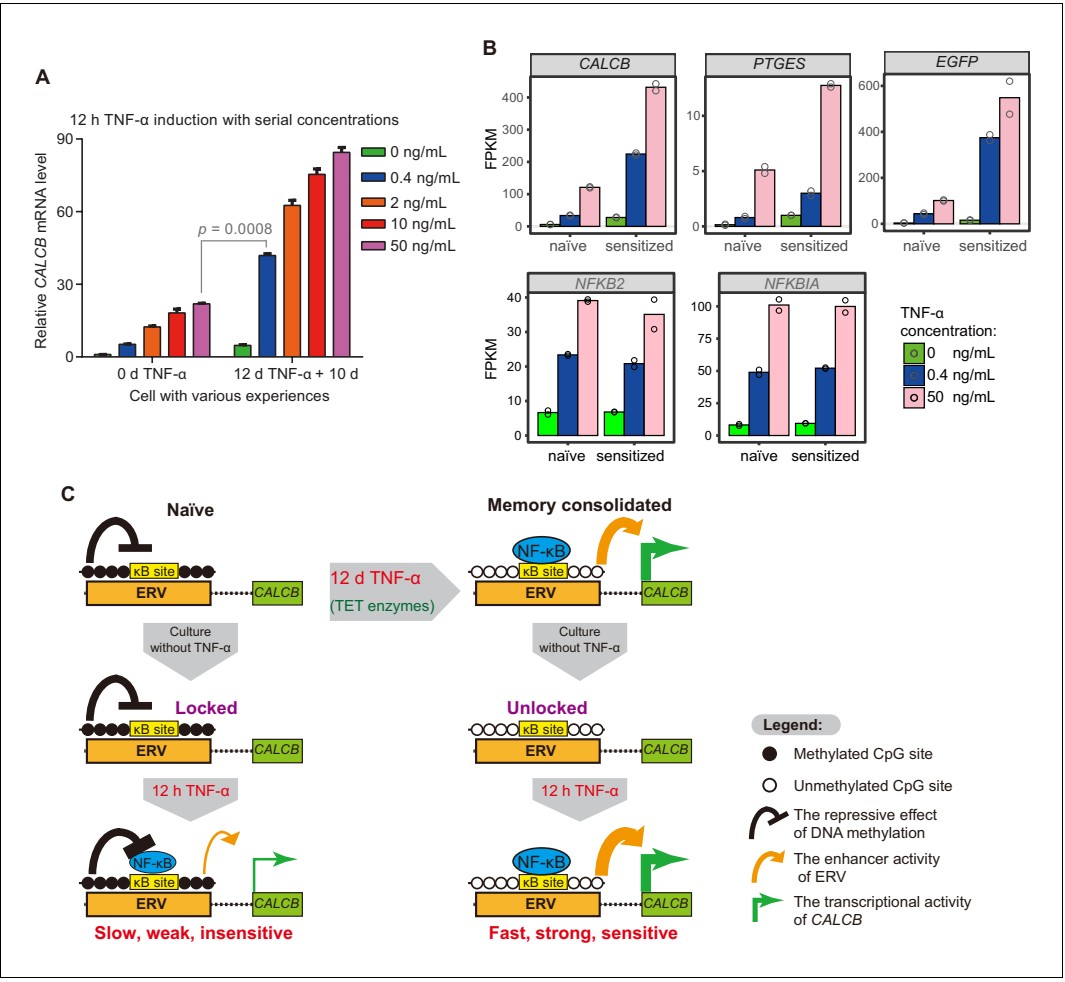

**Figure 7.** The memory consolidated cells are highly sensitive to low-dose TNF-α induction. (**A**) RT-qPCR results show the *CALCB* transcriptional level of naïve cells and of cells that had experienced 12 days of TNF-α treatment following 12 hr of 0.4 ng/mL, 2 ng/mL, 10 ng/mL, and 50 ng/mL TNF-α treatments. The cells were cultured in TNF-α-free media for 10 days before receiving a second TNF-α stimulation. *GAPDH* is used as the internal control. Data are shown as mean ± SD from two independent experiments. Two-tailed *t*-test. (**B**) Expression levels (in FPKM) of *EGFP*, *CALCB*, and *PTGES* for naïve cells and memory consolidated cells in response to 12 hr TNF-α induction with various TNF-α concentrations. *NFKB2* and *NFKBIA*, two TNF-α induced genes with no memory effect, are shown for comparison. Data are shown as the mean from two independent experiments. (**C**) Model illustration for the co-option of an ERV into the transcriptional memory module of *CALCB* in response to TNF-α. Memory consolidation offers faster, stronger, and more sensitive subsequent induction.

The online version of this article includes the following source data for figure 7:

**Source data 1.** Related to *Figure 7A*.
**Source data 2.** Related to *Figure 7B*.

with various concentrations of TNF-α (0.4 ng/mL, 2 ng/mL, 10 ng/mL, and 50 ng/mL) for 12 hr. Both cells displayed dose-dependent activation (*Figure 7A*). Amazingly, cells that had experienced prior TNF-α treatment could achieve an approximately twofold better effect with a 125-fold lower TNF-α concentration (0.4 ng/mL) than naïve cells treated with 50 ng/mL TNF-α (*Figure 7A*). To the best of our knowledge, this is the first transcriptional memory effect reported to increase signal sensitivity by more than 100-fold.

To confirm that cells could be sensitized by prior TNF-α treatment (50 ng/mL for 12 days), we performed RNA-seq experiments using the above cells with 0.4 ng/mL or 50 ng/mL TNF-α for a second induction of 12 hr and compared the results with those of naïve cells treated with 0.4 ng/mL or 50 ng/mL TNF-α for 12 hr. Indeed, *CALCB* and *EGFP* genes displayed even higher expression in cells with consolidated memory, despite having a 125-fold lower TNF-α signal (*Figure 7B*). In addition, *PEGES* reached a 60% expression level in memory consolidated cells treated with a 125-fold lower TNF-α signal (*Figure 7B*), indicating highly improved signal sensitivity.

## Discussion

This study has made several key discoveries with interesting implications for the fields of epigenetics and inflammation. From the epigenetics perspective, DNA demethylation appears to contribute to transcriptional memory events (*Murayama et al., 2006*; *Thomassin et al., 2001*), but the causal role of DNA demethylation in governing transcriptional memory has yet to be demonstrated. This study clearly demonstrated that TET enzymes are essential in governing transcriptional memory in response to TNF-α, providing a causal role for active DNA demethylation in consolidating transcriptional memory. From the inflammation perspective, a long-term puzzle is why acute inflammation can sometimes be transformed into chronic inflammation when the initial inflammatory signal is greatly reduced. Our discovery that sustained inflammatory signals can lead to the formation of transcriptional memory and that certain genes under the control of inflammatory transcriptional memory can even increase their sensitivity to inflammatory signals by 100-fold may provide some answers to this long-standing question.

### Systematic identification of transcriptional memory genes

Transcriptional memory has been studied for several decades, but the number of transcriptional memory genes remains relatively small. More examples need to be identified to reveal the significance of transcriptional memory. Two key difficulties exist in the identification of genes with transcriptional memory. First, they consist of only a small fraction of targeting genes responding to a particular signal of interest. Second, the duration of initial signal treatment can impact the process of memory consolidation. This study offers a pipeline to systematically identify genes with transcriptional memory by combining a defined signaling cue, a time course of the initial stimulus for memory consolidation, a dedicated transcriptome comparative analysis, and a functional validation with key regulator mutated cells. With this pipeline, we believe that many new transcriptional memory genes in response to various signaling cues will be quickly identified, which will certainly boost our understanding of the functional roles of transcriptional memory in many systems, such as the potential roles of transcriptional memory in immune cell memory.

### Signal-induced active DNA demethylation is likely a general principle for transcriptional memory

It is well known that transcription factor associations promote local demethylation (*Brandeis et al., 1994*; *Costa et al., 2013*; *de la Rica et al., 2013*; *Dubois-Chevalier et al., 2014*; *Fujiki et al., 2013*; *Kirillov et al., 1996*; *Macleod et al., 1994*; *Perera et al., 2015*; *Rampal et al., 2014*; *Sérandour et al., 2012*; *Silke et al., 1995*; *Tsai et al., 2014*; *Wang et al., 2015*; *Xiong et al., 2016*), and many transcription factors respond to various signaling cues. Our finding of inflammatory signal-induced DNA demethylation and its role in transcriptional memory consolidation is unlikely to be unique to inflammation signaling. We expect this to be an applicable principle for many other types of long-term transcriptional memory.

In theory, long-term transcriptional memory should be governed by epigenetic mechanisms that can be maintained for a long time in quiescent cells and can be faithfully inherited for many cell divisions in proliferating cells. DNA methylation is likely the best candidate because it meets both criteria. Importantly, CpG methylation tends to display a bimodal distribution (all or none) within a region, and this kind of digital behavior makes it an ideal candidate to store digital information at regulatory regions with a simple status code such as 'naïve/methylated/locked' and 'memory consolidated/unmethylated/unlocked' (*Figure 7C*). Whether this bimodal switch mediated by signal-induced demethylation also governs other types of transcriptional memory events is a highly interesting topic for future investigation and is relatively straightforward to test.

## High CpG density and initial methylation levels around the κB sites correlate with the functional potential to serve as inflammatory transcriptional memory modules

The methyl-CpG binding proteins are important for DNA methylation-mediated transcriptional repression (*Bird and Wolffe, 1999*). Moreover, the number and density of methylated cytosines are crucial for the efficiency of gene silencing (*Curradi et al., 2002*; *Hsieh, 1994*).

In this study, high CpG density and initial methylation level around the κB sites were found to positively correlate with the functional potential of these regions to serve as inflammatory transcriptional memory modules (*Figure 6H–J*), likely because DNA methylation is more effective in establishing a repressive environment in these regions, and sustained TNF-α induction is required to demethylate these regions to consolidate transcriptional memory.

## Methylation-insensitive TFs can be methylation-sensitive in the chromatin context and participate in transcriptional memory regulation

DNA methylation has a long-established role in gene silencing, but it is also known that gene activation can occur without demethylation (*Amedeo et al., 2000*; *Dong et al., 2018*; *Li et al., 2018a*). Some TFs contain CpG site(s) within their binding motifs and are sensitive to CpG methylation. Some TFs can bind and even favor binding motifs containing methylated CpG site(s). Therefore, TFs are classified as methylation-sensitive and methylation-insensitive TFs, respectively (*Tate and Bird, 1993*). Some other TFs, including NF-κB, are also methylation-insensitive when their binding motifs do not contain any CpGs. Indeed, p65 can activate methylated target genes without prior demethylation (*Zhao et al., 2019*; *Figures 3G* and *4D*). Therefore, the question is why transcriptional memory genes governed by a methylation-insensitive TF (NF-κB in this case) exhibit great expression differences in response to the same signal stimulus before and after DNA demethylation (*Figures 2E* and *7A*).

We believe that two explanations likely exist, and they may function independently or in combination to regulate transcriptional memory. First, as we presented, p65 exhibits marked preference toward the target sequence within unmethylated regions (*Figure 6C*), and demethylation lowers the transcriptional induction threshold and leads to higher gene induction with a 100-fold lower stimulus (*Figure 7A*). Second, methylation-insensitive, signal-dependent TFs may function together with methylation-sensitive TFs to regulate transcriptional memory events. In this context, DNA demethylation induced by the signal-dependent, methylation-insensitive TFs may facilitate the binding of other methylation-sensitive TFs and activate gene transcription in a coordinated manner.

The terms methylation-sensitive and methylation-insensitive TFs were originally defined by their interactions with naked DNA. We would like to emphasize that in the context of chromatin, even traditional methylation-insensitive TFs that contain no CpGs in their binding motifs greatly favor their binding sites embedded in unmethylated regions (*Figure 6C*).

## Binding is not functioning

κB sites embedded in regions with higher CpG density and initial methylation levels tended to display functional potential to serve as inflammatory transcriptional modules (*Figure 6H–J*); however, only a small fraction of these modules could actually drive the transcriptional memory events of adjacent genes (*Figures 2B* and *6N*). One possible explanation is that not all κB sites bound by p65 can drive the expression of neighboring genes. This again emphasizes the point that binding is not functioning, and transcriptional factor binding can only stimulate gene expression when enhancer-promoter connection is functionally wired.

## Migraine patients may suffer from the inflammatory transcriptional memory of *CALCB*

Migraine is a recurrent unilateral headache disorder (*Pellesi et al., 2017*), and it is one of the five leading causes of YLDs (*years lived with disability*) in the world, contributing to more than 30 million YLDs (*GBD 2016 Disease and Injury Incidence and Prevalence Collaborators, 2017*). CGRP overexpression is a major cause of migraine (*Edvinsson et al., 2018*; *Ho et al., 2010*; *Pellesi et al., 2017*; *Russell et al., 2014*; *Russo, 2015*; *Tepper, 2018*). Neurogenic neuroinflammation is proposed to be involved in the increase in migraine frequency, which leads to chronic migraine

(*Edvinsson et al., 2019*; *Malhotra, 2016*; *Ramachandran, 2018*). In this study, *CALCB* expression reached an FPKM of 699 after 12 days of TNF-α treatment (*Figure 2D*), and in memory consolidated cells, *CALCB* expression reached an FPKM of 225 after 12 hr with 125-fold lower TNF-α treatment (*Figure 7B*). These discoveries support a role for neuroinflammation in the pathogenesis of migraine.

The proinflammatory signal TNF-α induced gradual demethylation of *CALCB* enhancers within an ERV (*Figure 4C*), and memory consolidated cells encountering a subsequent TNF-α stimulus exhibited a robust elevation of *CALCB* expression (*Figure 2E*) and more than 100-fold sensitivity to TNF-α induction (*Figure 7A and B*). These features imply that inflammatory signals and memory consolidation likely play important pathogenic roles in migraine patients, especially in patients suffering from chronic migraine. Although these mechanisms remain highly speculative at this moment, this is a potentially interesting direction for clinical scientists studying migraine.

Additionally, although the *CALCB* gene is conserved between humans and rodents, we would like to point out that the ERV associated with *CALCB*, which regulates its inflammatory response, is primate-specific (*Figure 5E*). This suggests that it may be important to use nonhuman primate models to study whether the inflammation and inflammatory transcriptional memory of *CALCB* promote migraine pathogenesis.

# Materials and methods

## Key resources table

| Reagent type (species) or resource | Designation | Source or reference | Identifiers | Additional information |
|---|---|---|---|---|
| Cell line (*Homo sapiens*) | HEK293F with a stable reporter | PMID:29559556 | | |
| Cell line (*Homo sapiens*) | HEK293F with a stable reporter, *TET1* KO | PMID:30824537 | | |
| Cell line (*Homo sapiens*) | HEK293F with a stable reporter, *TET2* KO | PMID:30824537 | | |
| Cell line (*Homo sapiens*) | HEK293F with a stable reporter, *TET3* KO | PMID:30824537 | | |
| Cell line (*Homo sapiens*) | HEK293F with a stable reporter, *TET* TKO | PMID:30824537 | | |
| Cell line (*Homo sapiens*) | HEK293F with a stable reporter, *RELA* KO | PMID:30824537 | | |
| Cell line (*Homo sapiens*) | HEK293F with a stable reporter, MER11B-left KO | This paper | | Available upon request |
| Cell line (*Homo sapiens*) | HEK293F with a stable reporter, MER11B-right KO | This paper | | Available upon request |
| Cell line (*Homo sapiens*) | HEK293F with a stable reporter, MER11B-left+right KO | This paper | | Available upon request |
| Cell line (*Homo sapiens*) | HEK293F with a stable reporter, ERV KO | This paper | | Available upon request |
| Antibody | Rabbit anti-NF-κB p65 | Santa Cruz | Cat# sc-372; RRID:AB_632037 | 6 µL/ChIP |
| Antibody | Rabbit anti-H3K27ac | Active Motif | Cat# 39133; RRID:AB_2561016 | 2 µL/ChIP |
| Recombinant DNA reagent | lentiCRISPR v2 | Addgene | Cat# 52961 | |
| Peptide, recombinant protein | Recombinant human TNF-α | Peprotech | Cat# 300-01A | |
| Commercial assay or kit | KAPA HiFi Hotstart Uracil+ ReadyMix PCR kit | Kapa Biosystems | Cat# KK2801 | |
| Commercial assay or kit | CGRP (human) ELISA Kit | Cayman Chemical | Cat# 589101 | |

*Continued on next page*

*Continued*

| Reagent type (species) or resource | Designation | Source or reference | Identifiers | Additional information |
|---|---|---|---|---|
| Commercial assay or kit | EpiTect Bisulfite Kit | Qiagen | Cat# 59104 | |
| Commercial assay or kit | NEBNext Enzymatic Methyl-seq Conversion Module | NEB | Cat# E7125L | |
| Software, algorithm | BiQ Analyzer | *Bock et al., 2005* | | version 2.00 |
| Software, algorithm | FlowJo | Becton, Dickinson and Company | | version 7.6.1 |
| Software, algorithm | GraphPad Prism | GraphPad Software, La Jolla California USA | | version 7.00 |
| Software, algorithm | Bismark v0.22.3 | *Krueger and Andrews, 2011*; https://github.com/FelixKrueger/Bismark | PMID:21493656 | |
| Software, algorithm | Bowtie2 v2.3.5.1 | http://bowtie-bio.sourceforge.net/bowtie2/ | PMID:22388286 | |
| Software, algorithm | IGV v2.7.2 | http://software.broad institute.org/software/igv/ | PMID:21221095 | |
| Software, algorithm | Bedtools | Quinlan laboratory at the University of Utah | PMID:20110278 | |
| Software, algorithm | R software v3.6.0 | https://www.r-project.org/ | | The R Foundation |
| Software, algorithm | Rstudio v1.2.5042 | https://rstudio.com/ | | Rstudio Software |
| Software, algorithm | FastQC v0.11.9 | https://www.bioinformatics. babraham.ac.uk/projects/fastqc/ | | Simon Andrews |
| Software, algorithm | Cuffdiff v2.2.1 | *Trapnell et al., 2012*; http://cole-trapnell-lab.github.io/cufflinks/cuffdiff/ | PMID:22383036 | |
| Software, algorithm | STAR v2.7.3a | *Dobin et al., 2013*; https://github.com/alexdobin/STAR | PMID:23104886 | |
| Software, algorithm | DSS v2.34.0 | https://bioconductor.org/packages/DSS/ | PMID:26819470 | |
| Software, algorithm | deepTools v3.4.2 | *Ramírez et al., 2016*; https://github.com/deeptools/deepTools | PMID:27079975 | |
| Software, algorithm | Perl v5.26.2 | https://www.perl.org/ | | Perl-5.26.2 |

## Cell lines

All the HEK293F-derived cells are maintained in Dulbecco's Modified Eagle Medium (Gibco C11995500BT) with 10% fetal bovine serum (Biological Industries 04-010-1ACS 500 mL) and $1\times$ penicillin streptomycin solution (Sangon Biotech, China E607011-0100) using a cell culture incubator at 37°C and 5% $CO_2$. The HEK293F cells stably inserted with a DNA methylation silenced CMV reporter (*Figure 1A*) and its *TET1* KO, *TET2* KO, *TET3* KO, *TET* TKO, and *RELA* KO cells are previously established and described (*Li et al., 2018b*; *Zhao et al., 2019*). The MER11B-left KO, MER11B-right KO, MER11B-left+right KO, and ERV KO cells generated in this study are also derived from the HEK293F cells stably inserted with the DNA methylation silenced CMV reporter. The cell lines used in this study are free from mycoplasma contamination and authenticated: The HEK293F cells stably inserted with a DNA methylation silenced CMV reporter is established in Dr. Bing Zhu's lab and widely used in our previous studies (*Dong et al., 2018*; *Du et al., 2019*; *Li et al., 2018a*; *Li et al., 2018b*; *Zhao et al., 2019*); the flow cytometry results (*Figure 1C*) can confirm the insertion of DNA methylation-silenced CMV reporter.

## TNF-α treatment

The concentration of recombinant human TNF-α (Peprotech, USA 300-01A) for the treatment is 50 ng/mL unless otherwise stated. The schematic of TNF-α-mediated transcriptional memory

experiment is shown in *Figure 1B*. All the cultured HEK293F-derived cells are passaged every 2 days. For the long-term TNF-α induction, the proinflammatory cytokine TNF-α is added into the new culture medium immediately after cell passage.

## Measurement of CGRP release

For the CGRP release experiments, $3 \times 10^5$ cells were seeded per well in 6-well plates with 2 mL medium added and cultured for 2 days. Then the cell culture supernatants were collected to determine the levels of CGRP secretion. The Calcitonin Gene-Related Peptide (human) ELISA Kit (Cayman Chemical, USA 589101) and EnSight Multimode Plate Reader (PerkinElmer, USA) were used to measure CGRP concentration in the cell culture supernatants following the manufacturer's instructions.

## RT-qPCR for gene expression analysis

Cells are cultured and treated as described. Total RNA was extracted from cultured cells with TRIzol Reagent (Invitrogen, USA 15596026) following the manufacturer's instructions; 500 ng RNA was reverse-transcribed using HiScript II Q RT SuperMix for qPCR (+gDNA wiper) (Vazyme, China R223-01) according to the manufacturer's instructions. The synthesized cDNA was analyzed by quantitative real-time PCR using KAPA SYBR FAST Universal qPCR kit (Kapa Biosystems KK4601) and run on 7500 Fast Real-Time PCR system (Applied Biosystems, USA) with validated qPCR primers (*Supplementary file 1*). The gene expression changes were normalized to *GAPDH* transcript as an internal standard.

## ChIP-seq library preparation

Briefly, cells were fixed with 1% formaldehyde solution (Sigma, USA F1635) for 10 min at room temperature. The crosslinking reaction was stopped with glycine (0.125 M) and nuclei were prepared in cell lysis buffer (20 mM Tris-HCl pH 8.0, 85 mM KCl, 0.5% NP-40). Resuspend cell nuclei in nuclei lysis buffer (50 mM Tris-HCl pH 8.0, 10 mM EDTA-NaOH pH 8.0, 1% SDS, 1× complete EDTA-free protease inhibitor cocktail [Roche, Switzerland 04693132001]). Chromatin was sonicated using Covaris M220 Focused-ultrasonicator to an average length of approximately 250 bp. Cell debris was cleared by centrifugation and the supernatant was diluted 10-fold with dilution buffer (1% Triton X-100, 20 mM Tris-HCl pH 8.0, 200 mM NaCl, 2 mM EDTA-NaOH pH 8.0). Samples were incubated with anti-NF-κB p65 (C-20) rabbit polyclonal antibody (Santa Cruz Biotechnology, USA sc-372) overnight at 4°C. Antibody–chromatin complexes were pulled down with Protein A Dynabeads (Invitrogen 10002D) and were washed once with low salt wash buffer (0.1% SDS, 1% Triton X-100, 2 mM EDTA-NaOH pH 8.0, 20 mM Tris-HCl pH 8.0, 150 mM NaCl), twice with high salt wash buffer (0.1% SDS, 1% Triton X-100, 2 mM EDTA-NaOH pH 8.0, 20 mM Tris-HCl pH 8.0, 500 mM NaCl), once with LiCl wash buffer (0.25 M LiCl, 1% NP-40, 1% deoxycholate, 1 mM EDTA-NaOH pH 8.0, 10 mM Tris-HCl pH 8.0), and twice with TE buffer (10 mM Tris-HCl pH 8.0, 1 mM EDTA-NaOH pH 8.0). Incubate the beads with elution buffer (100 mM NaHCO$_3$, 1% SDS) to elute the complexes. Following crosslink reversal, RNase A and proteinase K treatment, phenol–chloroform extraction, and isopropanol precipitation, ChIP DNA samples were used to prepare libraries with Kapa hyper prep kit (Kapa Biosystems KK8504) and NEBNext multiplex oligos for Illumina (index primers set 1) (NEB, USA E7335) according to the manufacturer's protocol. Libraries were sequenced on an Illumina NovaSeq 6000 using 150 bp paired-end mode.

## ATAC-seq library preparation

The ATAC-seq libraries are prepared as previously described (*Corces et al., 2017*) with minor modifications. A total of 45,000 cells were collected and resuspended in 50 μL cold ATAC-Resuspension Buffer (10 mM Tris-HCl pH 7.4, 10 mM NaCl and 3 mM MgCl$_2$ in sterile water) containing 0.1% NP40, 0.1% Tween-20, and 0.01% Digitonin. Then the cells were incubated on ice for 3 min. The nuclei were washed with 1 mL cold ATAC-Resuspension Buffer containing 0.1% Tween-20 and centrifuged at 500 g and 4°C for 10 min. The cell pellets were resuspended in 50 μL transposition mix (10 μL 5 × TTBL, 5 μL TTE Mix V50, 16.5 μL PBS, 0.5 μL 1% digitonin, 0.5 μL 10% Tween-20, 17.5 μL ddH$_2$O) with TruePrep DNA Library Prep Kit V2 for Illumina (Vazyme, China TD501) and incubated at 37°C for 30 min with shaking in a thermomixer. The reactions were cleaned up with DNA Clean and Concentrator-5 (Zymo Research D4014). The libraries were constructed through PCR amplification

with TruePrep Index Kit V2 for Illumina (Vazyme, China TD202) to barcode samples. The ATAC-seq Libraries were sequenced on an Illumina NovaSeq 6000 (Berry Genomics Co., Ltd., China) using 150 bp paired-end mode.

## ChIP-seq and ATAC-seq analysis

ChIP-seq data for p65 and H3K27ac were filtered by trim_galore and aligned to human genome (hg38) using Bowtie2 software (v2.3.5.1). Potential PCR duplicates reads were removed and uniquely mapped reads were used for the subsequent analysis. Peaks were called using MACS2 (v2.1.2). p65 peaks in 12 hr TNF-α-treated naïve cells (first induction) and memory cells (second induction) were merged using Bedtools (v2.29.2) as consensus peak set, and peaks overlapping hg38 blacklist were removed. ATAC-seq data were processed in a similar way to ChIP-seq data. Aligned read pairs of ATAC-seq were deduplicated and used to generate genome coverage files in bigwig format with the normalization of CPM (counts per million). ChIP-seq and ATAC-seq profiles at regions of interest were drawn using 'plotProfile' tool in deepTools (v3.4.2). Gene track profiles were visualized using IGV (v2.7.2).

## Gene annotations and genomic features

Gene features of human genome hg38 were extracted from annotations in UCSC genome browser. Repeat elements were extracted from 'rmsk' table of UCSC genome browser and grouped by five classes. Nearest genes for each regulatory element were found using 'bedtools closest' tool according to the gene transcription start site annotation, and genes that were far than 50 kb were discarded.

## CRISPR-Cas9 system-mediated genome engineering

By using CRISPR-Cas9 system, gRNA sequences are designed to target the primate-specific ERV, MER11B-left element, MER11B-right element adjacent to *CALCB* for deletion (*Figure 5—figure supplements 1* and *2*). The gRNA sequences (*Supplementary file 1*) were cloned into lentiCRISPR v2 vectors (Addgene 52961) (*Sanjana et al., 2014*). HEK293F-derived cells are transfected with plasmids by lipofectamine 3000 transfection kit (Invitrogen L3000-015) according to the manufacturer's instructions. One and a half days after transfection, cells were placed under puromycin selection for 2 days. After one round of cell passage, isolation of clonal cells is achieved by serial dilutions in 96-well plates. Genotyping PCR with designed primers (*Supplementary file 1*) and Sanger sequencing are used for verification of individual clones to obtain the desired genome edited cell line.

## Loci-specific DNA methylation analysis

To perform the CMV promoter, MER11B elements (*Supplementary file 2*) loci-specific DNA methylation analysis, the purified genomic DNA was bisulfite converted with the EpiTect Bisulfite Kit (Qiagen, Germany 59104) following the manufacturer's instructions. The bisulfite-converted DNA was then amplified with Jumpstart REDTaq Readymix (Sigma P0982) using locus-specific PCR primers (*Supplementary file 1*) with nested PCR. The purified PCR products were cloned using the pEASY-T5 Zero Cloning Kit (TransGen Biotech, China CT501) and transformed into competent DH5α cells. Positive individual bacterial clones were selected and the inserted PCR products were sequenced by Sanger sequencing and analyzed by BiQ Analyzer (*Bock et al., 2005*). In the results of bisulfite sequencing, the individual sequenced clones are represented by the horizontal lines.

## Flow cytometry

To assess the GFP fluorescent intensity in the HEK293F-derived cells stably inserted with a *GFP* reporter gene, a single-cell suspension was prepared, sorted, and analyzed on BD FACSAria III (BD Biosciences) or BD FACSCalibur (BD Biosciences). The flow cytometry data was analyzed with FlowJo 7.6.1.

## Enzymatic Methyl-seq

The NEBNext Enzymatic Methyl-seq Conversion Module (NEB, USA E7125L) was used for measuring genome-wide DNA methylation level. Purified genomic DNA containing 0.05% CpG-methylated pUC19 control DNA and 1% unmethylated lambda DNA was sheared to a mean length of 450 bp

with Covaris M220 Focused-ultrasonicator. End repair, A-tailing, methylated adaptor (*Supplementary file 1*) ligation, and size selection were performed on 125 ng fragmented genomic DNA with Kapa hyper prep kit (Kapa Biosystems KK8504) according to the manufacturer's instructions. The methylated adaptor ligated DNA fragments were then treated with oxidation of 5mC and 5hmC, clean-up of TET2 converted DNA, denaturation of DNA, deamination of cytosines, and clean-up of deaminated DNA using the NEBNext Enzymatic Methyl-seq Conversion Module (NEB, USA E7125L). The deaminated DNA was amplified with KAPA HiFi Hotstart Uracil+ ReadyMix PCR kit using primers from NEBNext multiplex oligos for Illumina (index primers set 1; NEB E7335; Kapa Biosystems KK2801) according to the manufacturer's protocol. PCR cycling condition was as follows: 98° C for 45 s, followed by seven cycles of 98°C for 15 s, 60°C for 30 s, 72°C for 30 s, and final extension 72°C for 1 min, and then hold at 4°C. Libraries were sequenced on an Illumina NovaSeq 6000 using 150 bp paired-end mode.

### Genome-wide DNA methylation analysis

Enzymatic Methyl-seq reads in PE150 were first assessed by FASTQC software and then trimmed by trim_galore to remove adapters and low-quality bases. The filtered reads were then mapped to human genome (hg38) using the Bismark software (v0.22.3). Duplicate reads were discarded, and methylation information called by Bismark were then processed by 'DSS' (v2.34.0) (*Park and Wu, 2016*), a package in Bioconductor (v3.11), including smoothing, fetching average methylation of regions. Differential methylation around p65 peaks were calculated by CpG sites, based on the methylation information in 'Bismark' extracted data (cov $\geq$ 5), and plotted by the relative distance to κB motif (*Figure 4F*).

### Transcription analysis

All RNA-seq experiments were performed with two biological replicates for each sample by poly-A selection. RNA-seq reads were assessed by FastQC (v0.11.9) software and trimmed by trim_galore (v0.6.4) to remove adapters and low-quality bases. Based on the annotation of GENCODE Human Release 24 (GRCh38), the filtered reads were then mapped to human genome (hg38) using the STAR aligner (v2.7.3a). FPKM for genes were quantified using Cuffdiff (v2.2.1). The data of all protein coding genes were kept for the subsequent gene-based analysis. eRNA reads were quantified by counting RNA-seq reads that were mapped to p65-bound regulatory regions, normalized by sequencing depth and compared between samples.

### Quantification and statistical analysis

Statistical analyses were performed using Student's *t*-test in the R software or GraphPad Prism, two-tailed or one-tailed as indicated in the figure legends separately. Boxplots and barplots were drawn using R software.

## Acknowledgements

We thank Dr. Guohong Li for helpful discussions. We thank JY Jia and S Sun for technical assistance in FACS analysis. This work was supported by the Chinese Ministry of Science and Technology (2018YFE0203300), the NSFC-FDCT joint grant (31761163001 for BZ and 033/2017/AFJ for GL), the National Natural Science Foundation of China (31530047, 31761163001), and the Chinese Academy of Sciences (XDB 39000000 and QYZDY-SSW-SMC031). Z Zhang and JX are supported by the Youth Innovation Promotion Association (2017133 and 2020097, respectively) of the Chinese Academy of Sciences.

## Additional information

### Funding

| Funder | Grant reference number | Author |
| --- | --- | --- |
| Chinese Ministry of Science and Technology | 2018YFE0203300 | Bing Zhu |

| | | |
|---|---|---|
| National Natural Science Foundation of China | 31530047 | Bing Zhu |
| National Natural Science Foundation of China | 31761163001 | Bing Zhu |
| Chinese Academy of Sciences | XDB 39000000 | Bing Zhu |
| Chinese Academy of Sciences | QYZDY-SSW-SMC031 | Bing Zhu |
| Youth Innovation Promotion Association of the Chinese Academy of Sciences | 2017133 | Zhuqiang Zhang |
| Youth Innovation Promotion Association of the Chinese Academy of Sciences | 2020097 | Jun Xiong |
| National Natural Science Foundation of China | 31761163001 | Bing Zhu |
| Fundo para o Desenvolvimento das Ciências e da Tecnologia | 033/2017/AFJ | Gang Li |
| National Natural Science Foundation of China | 033/2017/AFJ | Gang Li |
| Fundo para o Desenvolvimento das Ciências e da Tecnologia | 31761163001 | Bing Zhu |

The funders had no role in study design, data collection and interpretation, or the decision to submit the work for publication.

### Author contributions

Zuodong Zhao, Conceptualization, Data curation, Formal analysis, Validation, Investigation, Visualization, Methodology, Writing - original draft, Project administration, Writing - review and editing, Z Zhao performed most of the experiments; Zhuqiang Zhang, Conceptualization, Data curation, Software, Funding acquisition, Investigation, Methodology, Writing - original draft, Project administration, Writing - review and editing, Z Zhang performed the bioinformatics analysis; Jingjing Li, Project administration, JL assisted in the experiments; Qiang Dong, Project administration, QD assisted in the experiments; Jun Xiong, Formal analysis, Funding acquisition, JX helped with data analysis; Yingfeng Li, Formal analysis, YL helped with data analysis; Mengying Lan, Project administration, M.L. assisted in the experiments; Gang Li, Funding acquisition, Investigation, G.L. participated in project design; Bing Zhu, Conceptualization, Data curation, Formal analysis, Supervision, Funding acquisition, Validation, Investigation, Writing - original draft, Writing - review and editing

### Author ORCIDs

Zuodong Zhao (ID) https://orcid.org/0000-0002-5858-4155
Zhuqiang Zhang (ID) https://orcid.org/0000-0001-6513-2854
Gang Li (ID) http://orcid.org/0000-0003-3203-8567
Bing Zhu (ID) https://orcid.org/0000-0003-2049-432X

### Decision letter and Author response

Decision letter https://doi.org/10.7554/eLife.61965.sa1
Author response https://doi.org/10.7554/eLife.61965.sa2

## Additional files

### Supplementary files

- Source code 1. The source code file for high-throughput data analysis in this study.
- Supplementary file 1. Sequence of the DNA oligonucleotides used in this study.
- Supplementary file 2. Genomic sequences for locus-specific bisulfite PCR sequencing analysis.

• Supplementary file 3. QC metrics for high-throughput experiments. Sheet 1: Basic QC stats for mRNA-seq experiments; Sheet 2: List of differentially expressed genes and FPKM values. Two replicates are shown; Sheet 3: Basic QC stats for WGBS experiments; Sheet 4: Differentially methylated regions (DMRs) between cells treated with 12 days TNF-α vs. 0 hr TNF-α cells; Sheet 5: DMRs between cells treated with 12 days TNF-α vs. 0 hr TNF-α *RELA* KO cells; Sheet 6: DMRs between cells treated with 12 days TNF-α vs. 0 hr TNF-α *TET2* KO cells; Sheet 7: Basic QC stats for ChIP-seq experiments; Sheet 8: Read counts of p65 peaks in various treated samples, normalized in reads per million per kb peak; Sheet 9: Basic QC stats for ATAC-seq experiments.

• Transparent reporting form

## Data availability

All high-throughput data generated in this study have been deposited in NCBI GEO database under accession number GSE152146, except that p65 ChIP-seq data for 0 h and 12 h TNF-α treatments have been deposited under the accession number GSE121361 (Zhao et al., 2019).

The following dataset was generated:

| Author(s) | Year | Dataset title | Dataset URL | Database and Identifier |
|---|---|---|---|---|
| Zhao Z, Zhang Z, Li J, Dong Q, Xiong J, Li Y, Lan M, Zhu B | 2020 | TNF induced inflammatory transcription dynamics and epigenetic changes | https://www.ncbi.nlm.nih.gov/geo/query/acc.cgi?acc=GSE152146 | NCBI Gene Expression Omnibus, GSE152146 |

The following previously published dataset was used:

| Author(s) | Year | Dataset title | Dataset URL | Database and Identifier |
|---|---|---|---|---|
| Zhao Z, Lan M, Li J, Dong Q, Li X, Liu B, Li G, Wang H, Zhang Z, Zhu B | 2019 | TNF-α induces IL-32 expression in HEK293 cells | https://www.ncbi.nlm.nih.gov/geo/query/acc.cgi?acc=GSE121361 | NCBI Gene Expression Omnibus, GSE121361 |

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
