## [Decision Letter]

**Acceptance summary:**

This study discovered inflammatory signal induced transcriptional memory, which is dependent on signal induced transcription factor activation and active DNA demethylation. Such transcriptional memory offers more rapid, more strong and more sensitive subsequent signal response. And this is likely a general principle that can be applied to other signaling systems.

**Decision letter after peer review:**

Thank you for submitting your article "Sustained TNF-α stimulation induces transcriptional memory that provides 100-fold more sensitive subsequent induction" for consideration by *eLife*. Your article has been reviewed by three peer reviewers, including Xiaobing Shi as the Reviewing Editor and Reviewer #1, and the evaluation has been overseen by Kevin Struhl as the Senior Editor.

The reviewers have discussed the reviews with one another and the Reviewing Editor has drafted this decision to help you prepare a revised submission.

Summary:

This manuscript explores the impact of DNA methylation on TNF-α-stimulated transcriptional memory. Transcriptional memory is a phenomenon that leads to heritable increase in the transcriptional response of certain genes to a stimulus that has been experienced previously. Transcriptional memory has been studied in many systems and has been shown to require transcription factors, co-activators, histone variants, histone modifications and physical interaction with nuclear pore proteins. The authors find that a GFP reporter under the control of a methylated NFkB-responsive CMV promoter exhibits a distinct form of transcriptional memory; after 10 days following previous exposure to TNF-α, the reporter shows both higher basal expression and greater responsiveness to TNF-α. This effect increases as the length of the previous treatment increases. Furthermore, the DNA methylation of the CMV promoter decreased over 12 days of TNF-α treatment. This loss of methylation and the increase in expression requires TET2 and, to some extent, TET3. From this, the authors propose that recruitment of TET2 (and perhaps TET3) to the CMV promoter during the initial induction demethylates the promoter, removing a heritable repressive mark and enhancing its responsiveness to TNF-α in the future. RNAseq experiments identified a small number of genes that behave like the CMV reporter. They focused on CALCB, a neuropeptide that stimulates vasodilation and has been implicated in migrane. TNF-α activates CALCB through NFkB and the level of expression increased over the course of 8 days of TNF-α treatment. Inactivation of TET2 reduced the accumulation for CALCB between 9h and 12d and these cells did not show memory. The authors find that two p65 binding sites near CALCB become demethylated during TNF-α treatment and remain less methylated for weeks. They propose that this promotes better binding by p65, leading to greater responsiveness. Finally, based on analysis of 10 p65 peaks near five genes that exhibit memory, the authors propose that highly methylated p65 binding sites with greater number of CpGs nearby are most affected by long-term treatment with TNF-α.

Overall, this is a very interesting study reporting a novel mechanism of TNF-α induced inflammatory transcriptional memory mediated by DNA demethylation. The involvement of DNA methylation in heritable changes in transcriptional regulation in response to environmental signals is of broad interest. The data is of high quality and generally supports the conclusions. However, there are a few concerns that need to be experimentally addressed to strengthen the paper.

1) The alternative mechanism to genomic/chromatin-based memory is that the physiological state of the cells or the activation status of the signaling pathways is different at the times of the first and second inductions. These possibilities need to be thoroughly tested.

2) It is possible that the loss of DNA methylation is due to non-specific effects of transcription on DNA methylation rather than specific demethylation. To distinguish between these possibilities, it is important to determine whether there is a direct protein-protein interaction between p65 and TET2, whether TET2 is recruited to p65 binding sites and if so, whether TET2 recruitment is dependent on p65.

3) The authors observed that TET deficient cells showed loss of transcription memory. Whether this loss of transcriptional memory is purely dependent on DNA methylation or requires p65 is not clear. P65 ChIP-seq or ChIP-qPCR at CALCB loci and / or CMV promoter in WT and TET-TKO cells will help to address this point.

4) Knock down TET2 after previously treating with TNF-α but before retreatment. If this disrupts memory (and leads to increased DNA methylation), it would argue that TET2-mediated demethylation is continuously important for proper inheritance. If it does not disrupt memory, it would argue that TET2-mediated demethylation occurs during the primary TNF-α treatment and that the low DNA methylation state is heritable afterward.

5) The expression of TET proteins and the 5hmC level are quite low in HEK293 cells. It is important to examine the endogenous TET protein and 5hmC levels in HEK293F cells used in this study during TNF-a stimulation. It would greatly strengthen the paper if the authors could validate the findings observed in this study in another system with relatively high TET expression, such as T cells, in which the epigenetic memory is important for cytokine production.

6) Does the memory feature apply to more genes than the few being examined in this study? The data suggesting a change in p65 occupancy or H3K27ac is primarily observed through averaging many sites and is less convincing for the sites that were studied functionally. The occupancy of p65 from the ChIP-seq experiments should be quantified and unbiased, statistical methods should be used to identify genes, in addition to the few being examined in this study, that have changed between the primary and secondary TNF-α treatment. Actually, for all the omics studies (RNA-seq, ChIP-seq, methyl-seq), it would be necessary to perform unbiased global analysis and provide the lists of differentially expressed gene (DEG), methylated sites, or enriched regions as part of QC. It seems that basic data QC information is missing in the supplementary materials. For example, how many reads were collected for each sample, what is the bisulfite conversion efficiency of methyl-seq, and what is the percentage of reads within identified peaks? All are essential for determining the qualify and rigor of these datasets.

---

## [Author Response]

Revisions for this paper:1) The alternative mechanism to genomic/chromatin-based memory is that the physiological state of the cells or the activation status of the signaling pathways is different at the times of the first and second inductions. These possibilities need to be thoroughly tested.

We thank the reviewers for the suggestion, and we agree it is an important question that should be clearly addressed.

First, we hierarchically clustered all differentially expressed genes (DEGs) during the two times of induction (Author response image 1 panel A). The naïve state cells (0 h) and memory state cells (12 d TNF-α + 10 d) have almost identical transcription profiles, and the DEGs generally show similar pattern between the first and the second inductions, 76 of 82 induced genes in the first treatment are also induced in the second treatment, with some genes exhibit higher induction during the second treatment (Author response image 1 panel A, B).

Then, by comparing the transcriptomes at naïve and memory states, we observe extremely high similarity between them (*r* = 0.999), except for a few genes with elevated baseline expression at the memory state (such as *CALCB* and *IL32*). The similar is true while comparing the 1^st^ and 2^nd^ inductions (*r* = 0.999), except that the memory genes now stand out. Based on the expression profiles, we believe that the general physiological state of the cells has not been changed after memory consolidation, except that a few memory genes become ready for a robust 2^nd^ induction. These results have been added as new Figure 2—figure supplement 1C. And we added the following paragraph into the manuscript:

Importantly, the transcriptomes between the naïve cells and memory consolidated cells were extremely similar (*r* = 0.999), and the same is true for the transcriptomes between first and second inductions (*r* = 0.999) (Figure 2—figure supplement 1C), indicating that the general physiology of the cells did not alter during memory consolidation, except for genes primed for memory response.

**Author response image 1. sa2fig1:** 

2) It is possible that the loss of DNA methylation is due to non-specific effects of transcription on DNA methylation rather than specific demethylation. To distinguish between these possibilities, it is important to determine whether there is a direct protein-protein interaction between p65 and TET2, whether TET2 is recruited to p65 binding sites and if so, whether TET2 recruitment is dependent on p65.

DNA demethylation induced by the binding of transcription factors has been reported in many examples (1-14). In certain examples, these transcription factors are reported to interact with the TET enzymes (6-11,13); While in some other cases, no direct evidence supporting the interaction between transcription factors and TET enzymes was provided (5,14). We expressed Flag-TET1, Flag-TET2, or Flag-TET3 in HEK293F cells and treated cells with 12 h TNF-α stimulation. Then we performed CoIP experiments between p65 and the TET enzymes, but we did not observe any convincing interactions even under low stringency washing conditions (150 mM NaCl) (data shown in Author response image 2). Therefore, we believe the TET enzymes do not directly interact with p65, although their demethylation functions are clearly dependent on p65 (Figure 4D).

On the other hand, increased chromatin accessibility has been reported to facilitate DNA demethylation mediated by the TET enzymes (15-18). To measure chromatin accessibility at p65 binding sites, we performed ATAC-seq experiments with cells under different treatments. Apparently, TNF-α treatment robustly increased local chromatin accessibility (Author response image 3). Taken together, we prefer a model that TET2-mediated demethylation is dependent on p65-induced chromatin environment changes but not direct protein-protein interaction between p65 and TET2.

**Author response image 3. sa2fig3:** 

To differentiate the contributions of p65 binding and p65-dependent transcriptional activation in TNF-α-induced demethylation, we determined p65 binding peaks in TNF-α-treated cells and selected those were heavily methylated in naïve cells, and then categorized them into three groups based on their eRNA expression levels. Interestingly, regardless whether eRNA can be detected from these p65 binding peaks, demethylation occurred in a similar way (new Figure 4H). Therefore, we conclude that p65 binding *per se*, but not active transcription is the main cause of TNF-α-induced demethylation. And we have highlighted these conclusions in our revised manuscript.

3) The authors observed that TET deficient cells showed loss of transcription memory. Whether this loss of transcriptional memory is purely dependent on DNA methylation or requires p65 is not clear. P65 ChIP-seq or ChIP-qPCR at CALCB loci and / or CMV promoter in WT and TET-TKO cells will help to address this point.

As suggested, p65 ChIP-qPCR experiments were performed for MER11B-left LTR upstream of *CALCB* in WT and *TET* TKO cells. And we observed significantly higher p65 occupancy at MER11B-left LTR in the second induction than the first induction for the WT cells, but not for the *TET* TKO cells (data shown in Author response image 4). These data support that the loss of transcriptional memory in *TET* TKO cells is dependent on the loss of demethylation that facilitates p65 binding.

**Author response image 4. sa2fig4:** 

4) Knock down TET2 after previously treating with TNF-α but before retreatment. If this disrupts memory (and leads to increased DNA methylation), it would argue that TET2-mediated demethylation is continuously important for proper inheritance. If it does not disrupt memory, it would argue that TET2-mediated demethylation occurs during the primary TNF-α treatment and that the low DNA methylation state is heritable afterward.

We attempted to perform this experiment and constructed 4 different shRNA vectors targeting *TET2*, unfortunately the knock down efficiency was not satisfactory. Ideally, we can generate an inducible *TET2* knockout and deletion it after memory consolidation. After careful assessment, we have decided not to perform this experiment for two reasons and we hope the reviewer will agree with our choice. First, inducible *TET2* knockout is a complicated strategy that requires a substantial amount of time to perform and will delay the publication. Moreover, DNA methylation maintenance is very robust and DNA methylation pattern is highly stable in most somatic cells. We have shown that in the absence of TNF-α treatment (no p65 binding and p65-dependent TET2 function) for 30 days, which equals approximately 30 cell divisions, *CALCB* transcriptional memory remained unchanged (Figure 2F) and *CALCB* enhancers remained unmethylated (Figure 4—figure supplement 1D). These results suggest the likelihood of de novo methylation at these regions and the requirement of TET2 in preventing such de novo methylation are quite low.

5) The expression of TET proteins and the 5hmC level are quite low in HEK293 cells. It is important to examine the endogenous TET protein and 5hmC levels in HEK293F cells used in this study during TNF-a stimulation. It would greatly strengthen the paper if the authors could validate the findings observed in this study in another system with relatively high TET expression, such as T cells, in which the epigenetic memory is important for cytokine production.

The FPKM values of TET proteins in RNA-seq are listed in Author response table 1. Although not very high, all three TET proteins are stably expressed.

**Author response table 1. resptable1:** 

Treatment	TET1	TET2	TET3
0 h	2.31	2.97	6.05
12 h TNF-α	2.33	2.90	9.29
12 d TNF-α	3.04	2.77	6.89
12 d TNF-α + 10 d	2.05	2.79	5.92
12 d TNF-α + 10 d + 12 h TNF-α	2.47	3.29	9.97

We also used UHPLC-MRM MS/MS to quantify the 5hmC levels in the genomic DNA extracted from HEK293F cells, with various time courses of TNF-α induction. (listed in Author response table 2). 5hmC can be stably detected in all samples, indicating the presence of a functioning TET enzyme system in these cells.

**Author response table 2. resptable2:** 

TNF-α Treatment	5hmdC/10^6^ dC	SD
0 d	22.6	1.2
2 d	30.2	1.3
4 d	31.2	0.6
6 d	32.2	1.9
8 d	37.8	4.2
12 d	37.6	1.6

5hmdC indicates 5-hydroxymethyl-2’-deoxycytidine.

dC indicates 2’-deoxycytidine. SD represents standard deviation.

We agree with reviewers that it is important to study TF-induced, TET2-dependent transcriptional memory in other interesting systems. Indeed, we are currently actively working on this in B cell memory system. We would love to see the mechanism that we observe in a model cell type can be applied to more interesting physiological systems, but we think these can be an entirely independent piece of study.

6) Does the memory feature apply to more genes than the few being examined in this study? The data suggesting a change in p65 occupancy or H3K27ac is primarily observed through averaging many sites and is less convincing for the sites that were studied functionally.

We agree with the reviewers’ judgement, and we think in this model cell type, the number of real endogenous memory genes is limited. Like point 5 raised by the reviewers and our response, we believe in other systems, we may observe much more profound signal-induced transcriptional memory genes, and we would love to prove that in our next study.

Regarding why the functional significance of memory modules appear to be more modest than the chromatin feature indicators, we would like to quote a sentence often mentioned by Dr. Robert Roeder and others “Binding is functioning”. To us, p65 occupancy, or ATAC-seq signal or K27ac signal are good indications that these regions have the functional potential to be enhancers. However, enhancer-promoter wiring is another critical issue to make these potentially functioning units into real functioning units. This process involves many other factors and that is why only a fraction of these potential enhancers can truly activate genes. But based on the eRNA expression, these potential enhancers are already idling. They just do not have the opportunity to be connected with a promoter in this particular cell type. We have a paragraph in the Discussion section discussing this.

The occupancy of p65 from the ChIP-seq experiments should be quantified and unbiased, statistical methods should be used to identify genes, in addition to the few being examined in this study, that have changed between the primary and secondary TNF-α treatment.

As suggested, we systematically analyzed the occupancy of p65 (new Figure 6—figure supplement 1), and found that occupancy-increased peaks are prone to highly methylated and demethylated after 12 d TNF-α treatment (new Figure 6G). In another direction, we also compared p65 occupancy, ATAC-seq signal, and H3K27ac signal according to various level of total methylated CpGs, and apparently the reduction of total CpGs methylation promoted p65 binding, chromatin opening and H3K27ac (new Figure 6H-J).

Actually, for all the omics studies (RNA-seq, ChIP-seq, methyl-seq), it would be necessary to perform unbiased global analysis and provide the lists of differentially expressed gene (DEG), methylated sites, or enriched regions as part of QC. It seems that basic data QC information is missing in the supplementary materials. For example, how many reads were collected for each sample, what is the bisulfite conversion efficiency of methyl-seq, and what is the percentage of reads within identified peaks? All are essential for determining the qualify and rigor of these datasets.

According to reviewers’ suggestion, all DEGs, differentially methylated sites and other basic QC information are listed in the Supplementary file 3.

**References**

1. Brandeis M, Frank D, Keshet I, Siegfried Z, Mendelsohn M, Nemes A, Temper V, Razin A, and Cedar H. (1994) Sp1 elements protect a CpG island from de novo methylation. *Nature* 371, 435-4382.

2. Macleod D, Charlton J, Mullins J, and Bird AP. (1994) Sp1 sites in the mouse aprt gene promoter are required to prevent methylation of the CpG island. *Genes & development* 8, 2282-22923.

3. Silke J, Rother KI, Georgiev O, Schaffner W, and Matsuo K. (1995) Complex demethylation patterns at Sp1 binding sites in F9 embryonal carcinoma cells. *FEBS Letters* 370, 170-1744.

4. Kirillov A, Kistler B, Mostoslavsky R, Cedar H, Wirth T, and Bergman Y. (1996) A role for nuclear NF-kappaB in B-cell-specific demethylation of the Igkappa locus. *Nature Genetics* 13, 435-4415.

5. Serandour AA, Avner S, Oger F, Bizot M, Percevault F, Lucchetti-Miganeh C, Palierne G, Gheeraert C, Barloy-Hubler F, Peron CL, Madigou T, Durand E, Froguel P, Staels B, Lefebvre P, Metivier R, Eeckhoute J, and Salbert G. (2012) Dynamic hydroxymethylation of deoxyribonucleic acid marks differentiation-associated enhancers. *Nucleic Acids Research* 40, 8255-82656.

6. Costa Y, Ding J, Theunissen TW, Faiola F, Hore TA, Shliaha PV, Fidalgo M, Saunders A, Lawrence M, Dietmann S, Das S, Levasseur DN, Li Z, Xu M, Reik W, Silva JC, and Wang J. (2013) NANOG-dependent function of TET1 and TET2 in establishment of pluripotency. *Nature* 495, 370-3747.

7. de la Rica L, Rodriguez-Ubreva J, Garcia M, Islam AB, Urquiza JM, Hernando H, Christensen J, Helin K, Gomez-Vaquero C, and Ballestar E. (2013) PU.1 target genes undergo Tet2-coupled demethylation and DNMT3b-mediated methylation in monocyte-to-osteoclast differentiation. *Genome Biology* 14, R998.

8. Fujiki, K., Shinoda, A., Kano, F., Sato, R., Shirahige, K., and Murata, M. (2013) PPARgamma-induced PARylation promotes local DNA demethylation by production of 5-hydroxymethylcytosine. *Nature Communications* 4, 22629.

9. Dubois-Chevalier J, Oger F, Dehondt H, Firmin FF, Gheeraert C, Staels B, Lefebvre P, and Eeckhoute J. (2014) A dynamic CTCF chromatin binding landscape promotes DNA hydroxymethylation and transcriptional induction of adipocyte differentiation. *Nucleic Acids Research* 42, 10943-1095910.

10. Rampal R, Alkalin A, Madzo J, Vasanthakumar A, Pronier E, Patel J, Li Y, Ahn J, Abdel-Wahab O, Shih A, Lu C, Ward PS, Tsai JJ, Hricik T, Tosello V, Tallman JE, Zhao X, Daniels D, Dai Q, Ciminio L, Aifantis I, He C, Fuks F, Tallman MS, Ferrando A, Nimer S, Paietta E, Thompson CB, Licht JD, Mason CE, Godley LA, Melnick A, Figueroa ME, and Levine RL. (2014) DNA hydroxymethylation profiling reveals that WT1 mutations result in loss of TET2 function in acute myeloid leukemia. *Cell Reports* 9, 1841-185511.

11. Tsai YP, Chen HF, Chen SY, Cheng WC, Wang HW, Shen ZJ, Song C, Teng SC, He C, and Wu KJ. (2014) TET1 regulates hypoxia-induced epithelial-mesenchymal transition by acting as a co-activator. *Genome biology* 15, 51312.

12. Perera A, Eisen D, Wagner M, Laube SK, Kunzel AF, Koch S, Steinbacher J, Schulze E, Splith V, Mittermeier N, Muller M, Biel M, Carell T, and Michalakis S. (2015) TET3 is recruited by REST for context-specific hydroxymethylation and induction of gene expression. *Cell Reports* 11, 283-29413.

13. Wang Y, Xiao M, Chen X, Chen L, Xu Y, Lv L, Wang P, Yang H, Ma S, Lin H, Jiao B, Ren R, Ye D, Guan KL, and Xiong Y. (2015) WT1 recruits TET2 to regulate its target gene expression and suppress leukemia cell proliferation. *Molecular Cell* 57, 662-67314.

14. Xiong J, Zhang Z, Chen J, Huang H, Xu Y, Ding X, Zheng Y, Nishinakamura R, Xu GL, Wang H, Chen S, Gao S, and Zhu B. (2016) Cooperative Action between SALL4A and TET Proteins in Stepwise Oxidation of 5-Methylcytosine. *Molecular Cell* 64, 913-92515.

15. Shen L, Wu H, Diep D, Yamaguchi S, D'Alessio AC, Fung HL, Zhang K, and Zhang Y. (2013) Genome-wide analysis reveals TET- and TDG-dependent 5-methylcytosine oxidation dynamics. *Cell* 153, 692-70616.

16. Wu H, Wu X, Shen L, and Zhang Y. (2014) Single-base resolution analysis of active DNA demethylation using methylase-assisted bisulfite sequencing. *Nature Biotechnology* 32, 1231-124017.

17. Sun Z, Dai N, Borgaro JG, Quimby A, Sun D, Correa IR, Jr, Zheng Y, Zhu Z, and Guan S. (2015) A sensitive approach to map genome-wide 5-hydroxymethylcytosine and 5-formylcytosine at single-base resolution. *Molecular Cell* 57, 750-76118.

18. Xia B, Han D, Lu X, Sun Z, Zhou A, Yin Q, Zeng H, Liu M, Jiang X, Xie W, He C, and Yi C. (2015) Bisulfite-free, base-resolution analysis of 5-formylcytosine at the genome scale. *Nature Methods* 12, 1047-1050